# Modelling POLG mutations in mice unravels a critical role of POLγB in regulating phenotypic severity

Samantha Corrà[1,14], Alessandro Zuppardo [1,2,14], Sebastian Valenzuela [3,14], Louise Jenninger[3,14], Raffaele Cerutti[1,2], Sirelin Sillamaa [3], Emily Hoberg[3], Katarina A. S. Johansson [3], Urska Rovsnik[3], Sara Volta[1,2], Pedro Silva-Pinheiro [4], Hannah Davis[5], Aleksandra Trifunovic [6], Michal Minczuk [4,7], Claes M. Gustafsson [3], Anu Suomalainen [8,9], Massimo Zeviani[10,11], Bertil Macao [3], Xuefeng Zhu [3,12,13] ✉, Maria Falkenberg [3] ✉ & Carlo Viscomi [1,2] ✉

DNA polymerase γ (POLγ), responsible for mitochondrial DNA replication, consists of a catalytic POLγA subunit and two accessory POLγB subunits. Mutations in *POLG*, which encodes POLγA, lead to various mitochondrial diseases. We investigated the most common *POLG* mutations (A467T, W748S, G848S, Y955C) by characterizing human and mouse POLγ variants. Our data reveal that these mutations significantly impair POLγ activities, with mouse variants exhibiting milder defects. Cryogenic electron microscopy highlighted structural differences between human and mouse POLγ, particularly in the POLγB subunit, which may explain the higher activity of mouse POLγ and the reduced severity of mutations in mice. We further generated a panel of mouse models mirroring common human *POLG* mutations, providing crucial insights into the pathogenesis of *POLG*-related disorders and establishing robust models for therapeutic development. Our findings emphasize the importance of POLγB in modulating the severity of *POLG* mutations.

DNA polymerase γ (POLγ), the sole polymerase for mitochondrial DNA (mtDNA) replication, functions as a heterotrimer with one catalytic POLγA subunit and two POLγB accessory subunits[1]. The *POLG* gene encodes the 140 kDa POLγA subunit, which possesses polymerase, exonuclease, and lyase activities, while *POLG2* encodes the 55 kDa POLγB subunit, which enhances DNA binding and processivity[1]. In addition, POLγ activity requires the TWINKLE DNA helicase[2] and the mitochondrial single-stranded DNA-binding protein (mtSSB), which

[1]Veneto Institute of Molecular Medicine (VIMM), Via Orus, 2-35129 Padova, Italy. [2]Department of Biomedical Sciences, University of Padova, Via Ugo Bassi, 58/B-35131 Padova, Italy. [3]Department of Medical Biochemistry and Cell Biology, University of Gothenburg, Medicinaregatan 9A, P.O. Box 440, 41390 Gothenburg, Sweden. [4]MRC Mitochondrial Biology Unit, University of Cambridge, Hills Road, Cambridge CB2 0XY, UK. [5]The Mary Lyon Centre, MRC Harwell, Becquerel Ave, Didcot, Oxfordshire OX11 0RD, UK. [6]Institute for Mitochondrial Diseases and Aging, Faculty of Medicine, CECAD Research Center, 50931 Cologne, Germany. [7]Department of Clinical Neurosciences, University of Cambridge, Cambridge CB2 2PY, UK. [8]Research Programs Unit, Stem Cells and Metabolism, Biomedicum-Helsinki, Haartmaninkatu 8, University of Helsinki, 00290 Helsinki, Finland. [9]HUSlab, Helsinki University Hospital, University of Helsinki, 00290 Helsinki, Finland. [10]Department of Neurosciences, University of Padova, Via Belzoni 160, 35121 Padova, Italy. [11]Institute for Maternal and Child Health, IRCCS "Burlo Garofolo", Via Istria 61, 34137 Trieste, Italy. [12]School of Basic Medical Sciences, North China University of Science and Technology, Tangshan 063210, China. [13]National Key Laboratory for Development and Utilization of Forest Food Resources, Zhejiang A&F University, Hangzhou 311300, China. [14]These authors contributed equally: Samantha Corrà, Alessandro Zuppardo, Sebastian Valenzuela, Louise Jenninger. ✉e-mail: xuefeng.zhu@gu.se; maria.falkenberg@medkem.gu.se; carlo.viscomi@unipd.it

stimulates mtDNA synthesis by increasing both TWINKLE helicase and POLγ polymerase activities[3].

With over 300 known variants, *POLG* mutations are the leading cause of mitochondrial disease linked to a single nuclear gene[4]. The A467T, W748S, G848S, and T251I–P587L recessive mutations, along with the Y955C dominant mutation, are the most frequent amino acid changes found in patients[5]. The clinical phenotypes associated with POLγ impairment are highly heterogeneous and include Alpers-Huttenlocher encephalopathy (AHS), ataxia syndromes such as mitochondrial recessive ataxia syndrome (MIRAS) and sensory ataxia neuropathy, dysarthria, and ophthalmoplegia (SANDO), and autosomal dominant or autosomal recessive progressive external ophthalmoplegia (PEO)[6]. These diseases manifest across a broad age range, from infancy to late adulthood[7–9], with genotype-phenotype correlations often proving elusive. An additional factor potentially contributing to the variability of *POLG*-related diseases is the role of POLγ in regulating the immune response to viral infections[10].

The recessive W748S mutation belongs to the Finnish disease heritage, and both A467T and W748S occur at a frequency of approximately 1 in 50 in Northern European populations[10]. These mutations affect different regions of the POLγA protein: A467T is located in the thumb region of the polymerase domain, while W748S resides in the linker region, which connects the proofreading and polymerase domains and plays a crucial role in POLγA–POLγB interactions. The precise mechanisms by which individual mutations impair POLγ activity and lead to disease remain incompletely understood. In vitro studies indicate that the A467T mutation reduces POLγA's affinity for the POLγB accessory subunit and impairs its catalytic activity[11]. W748S, which is commonly associated with the E1143G polymorphism, causes impaired DNA polymerase activity, low processivity, reduced DNA affinity, and slightly weakened interactions with POLγB[11,12]. The G848S mutation dramatically reduces polymerase activity and decreases DNA affinity compared to the wild-type enzyme[13,14].

The Y955C mutation affects the catalytic site of POLγA, impairing its interaction with the incoming nucleotide and reducing dNTP incorporation without altering DNA affinity[15–19]. Consequently, the POLγ[Y955C] holoenzyme is approximately twofold less accurate than POLγ[WT], despite having a functional 3′–5′-exonuclease domain[15,20]. Interestingly, increasing dNTP concentrations can partially rescue the DNA synthesis activity of the POLγ[Y955C]-variant[14].

The three most common recessive mutations, A467T, W748S, and G848S, have been observed in various combinations (https://tools.niehs.nih.gov/polg/). These include compound heterozygous combinations such as A467T/W748S, A467T/G848S, and W748S/G848S, as well as homozygous states (A467T/A467T and W748S/W748S). Notably, while homozygous mutations can still result in severe symptoms, they are generally less severe than compound heterozygous combinations, except for G848S in the homozygous state, which has been identified in only four patients, all of whom died within 2 months from birth[21,22]. Additionally, compound heterozygous combinations involving G848S are associated with more severe phenotypes. In contrast, dominant mutations such as Y955C primarily lead to autosomal dominant PEO. To date, no effective therapeutic strategies exist to treat or cure these severe, progressive disorders.

Polγ mouse models are crucial for deciphering disease mechanisms and informing therapeutic development. PolγA and PolγB are conserved in most mammals, with mPolγA sharing 85% sequence identity with its human counterpart[23]. We have previously reported the generation of two mouse models (*Polg[A449T/A449T]* and *Polg[W726S/W726S]*), which replicate the human A467T and W748S mutations and recapitulate aspects of the human clinical phenotypes[10,11]. However, more comprehensive studies on disease-causing *POLG* mutations in mouse models are critically needed to understand the complexity of the human syndromes. In particular, dominant and compound heterozygous mutations commonly observed in affected patients remain poorly investigated.

Here, we establish a panel of mouse models replicating the most common human *POLG* mutations and integrate in vitro, biochemical, and structural analyses to reveal key genotype-activity-phenotype relationships. These models serve as a foundation for mechanistic insights and therapeutic exploration in *POLG*-related diseases.

## Results

### Disease-causing variants affect both human and mouse POLγ activity

To investigate the molecular mechanisms underlying POLG disorders and compare the activities of human and mouse POLγ, we expressed and purified wild-type human (hPOLγ[WT]) and mouse (mPolγ[WT]) polymerase, as well as the three most common recessive disease mutants, hPOLγ[A467T], hPOLγ[W748S], and hPOLγ[G848S], along with their corresponding mouse variants: mPolγ[A449T], mPolγ[W726S], and mPolγ[G826S]. Additionally, we analyzed the most common dominant mutation, hPOLγ[Y955C], and its corresponding mouse variant, mPolγ[Y933C]. All proteins were expressed in recombinant form, and their concentrations were determined before enzymatic assays (Supplementary Fig. 1a). Unless otherwise stated, all wild-type and mutant variants (both human and mouse) were used at equal concentrations in the biochemical experiments.

To assess the effect of the *POLG* mutations on polymerase activity, we annealed a radiolabeled primer to circular, single-stranded M13mp18 DNA, providing a pre-existing 3′-OH group for POLγ to initiate DNA synthesis (Fig. 1a). Both hPOLγ[WT] and mPolγ[WT] were active across a broad range of dNTP concentrations, producing full-length DNA products within 20 min at dNTP concentrations of 1 μM and higher (Fig. 1b). In contrast, all hPOLγ and mPolγ mutant variants displayed reduced activity. Interestingly, all mouse variants exhibited higher activity than their human counterparts (Fig. 1b–d).

To minimize variability in our experiments, we consistently used human mtSSB and only varied POLγ between its human and mouse variants. To validate this approach, we repeated the experiment using mouse mtSSB, obtaining similar results (compare Fig. 1b, c with Supplementary Fig. 1b, c).

We also tested the dominant hPOLγ[Y955C] and mPolγ[Y933C] mutations at higher dNTP concentrations (250 and 500 μM), given their known reduction in dNTP affinity[14]. Even at these extremely high, non-physiological dNTP concentrations, the mutants remained less active than the wild type (Fig. 1d).

To obtain quantitative data on the differences, we measured the effect of each individual mutation relative to the wild-type enzyme in both human and mouse systems. These experiments were performed at physiologically relevant dNTP concentrations (10 μM) and low protein concentrations to ensure that the reactions remained non-saturated. We found that while the relative effect of each mutation followed the same trend in both species, all mutations had a milder impact on polymerase activity in the mouse system compared to the human system. Most notably, the mPolγ[A449T] variant, exhibited unexpectedly high activity, surpassing the predicted effect based on its human counterpart, hPOLγ[A467T] (compare Fig. 1e, f).

### Impact of POLγ mutations on dsDNA replication efficiency

In vivo, POLγ requires the TWINKLE DNA helicase and mtSSB to form a minimal replisome that functions on double-stranded DNA (dsDNA). TWINKLE unwinds the dsDNA ahead of POLγ, with mtSSB stimulating the reaction. To assess the effects of mutant POLγ variants in the presence of these replication factors, we used a template containing a ~4 kb dsDNA region with a preformed replication fork (Fig. 2a). In a rolling circle replication assay, we measured DNA synthesis over time by incorporating radioactive dNTPs. To minimize variability, human mtSSB and TWINKLE were used in all experiments, with only POLγ varied between its human and mouse variants.

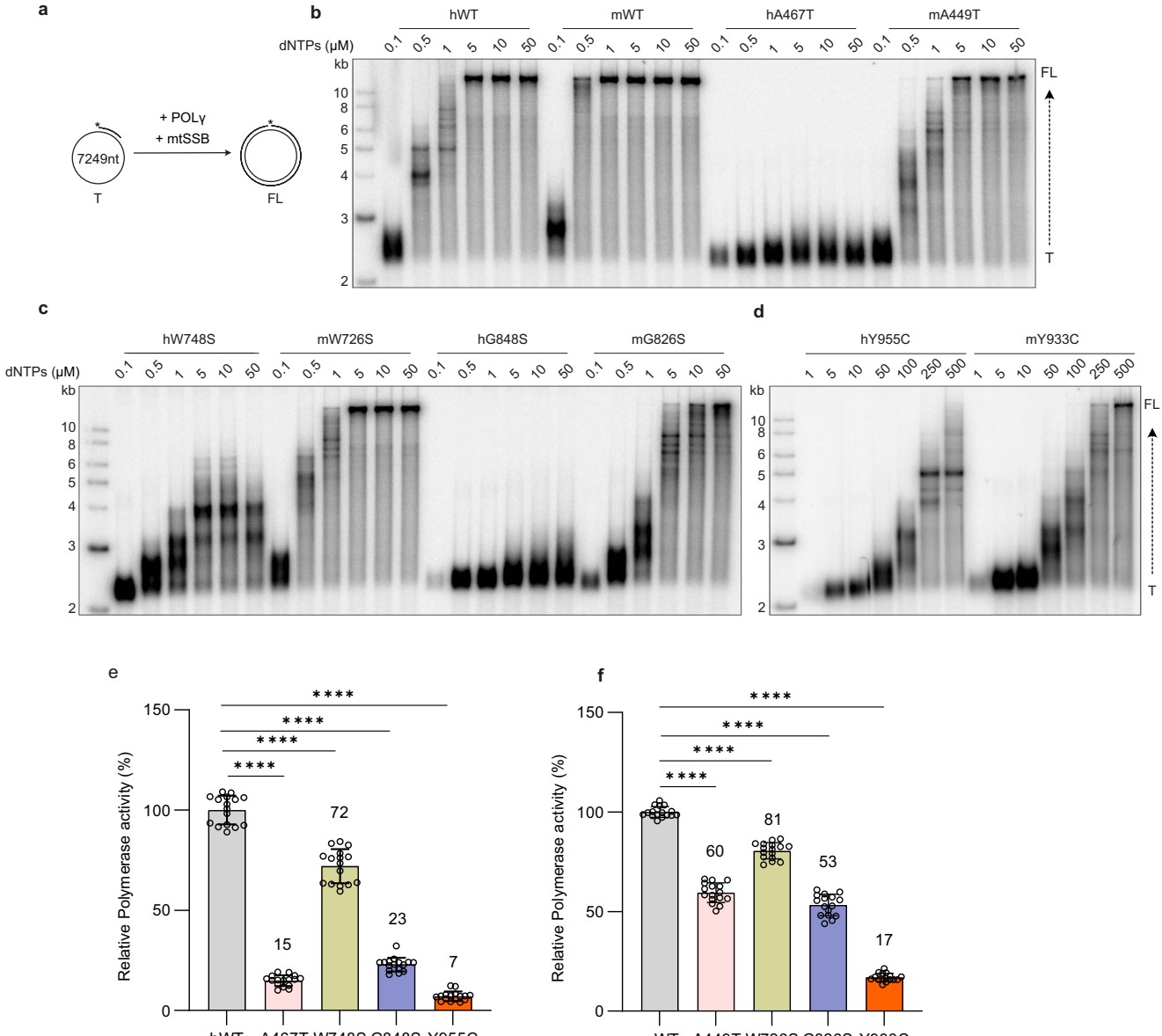

**Fig. 1 | Mutations impair polymerase activity across all tested dNTP concentrations. a** Schematic representation of the in vitro replication assay on a single-stranded template. A short 20-nt primer was annealed to circular, single-stranded M13mp18 DNA, providing a pre-existing 3'-OH group from which POLγ could initiate DNA synthesis and proceed until synthesizing the full molecule. **b–d** Both hPOLγWT and mPolγWT, as well as their mutant variants, were tested across a broad range of dNTP concentrations. Polymerase and human mtSSB were incubated with a primed, single-stranded DNA template at 37 °C for 20 min and analyzed on a 0.8% native agarose gel. The positions of the full-length (FL) product and the radiolabeled primer template (T) are indicated. All mutant variants exhibited reduced activity at all tested dNTP concentrations compared to their respective WT counterparts. $n = 3$ independent experiments. **e** Comparison of DNA

synthesis activity between hPOLγ[WT] and disease-causing mutations in hPOLγ. The assay was performed at lower Polymerase concentrations and with 10 μM dNTPs to ensure synthesis remained in the linear phase. DNA synthesis activity was assessed using the template described in (**a**) with an unlabeled primer. Polymerase activity was measured after 60 min at 37 °C by monitoring SYBR Green fluorescence, indicative of newly formed double-stranded DNA, enabling comparison between the variants. **f** Comparison of DNA synthesis activity between mPoly variants and mPoly[WT]. The experimental conditions were similar to (**e**). Relative activity to corresponding wild type (Mean, %) indicated above the bar. The error bars present the standard deviation and $n = 16$ technical replicates in (**e**, **f**). For multiple comparisons in **e** and **f**, Welch and Brown-Forsythe one-way ANOVA was used to determine $p$ values. ****$p < 0.0001$. Source data are provided as a Source Data file.

The wild-type replication machinery, reconstituted with recombinant human or mouse POLγ, efficiently supported DNA synthesis, producing products up to 20 kb or more in length (Fig. 2b and Supplementary Fig. 2a). We then assessed the effects of mutant POLγ variants on rolling-circle replication (Fig. 2b and Supplementary Fig. 2a). All three, recessive human mutations impaired activity, with hPOLγ[G848S] showing the strongest reduction, consistent with its more severe effect in affected individuals compared to the other mutations. The corresponding mouse mutations, mPoly[A449T],

mPolγ[W726S], and mPolγ[G826S], also impaired rolling-circle replication, but the effects were less pronounced than those observed with their human counterparts.

To monitor effects of the recessive mutations in the presence of POLγ[WT], a situation that reflects heterozygous carriers, we mixed wild-type and mutant variants at a 1:1 molar ratio. The concentration of each individual protein was reduced by half so that the total concentration remained the same as in experiments where only one variant was used. Under these conditions, we observed generally mild decreases in DNA

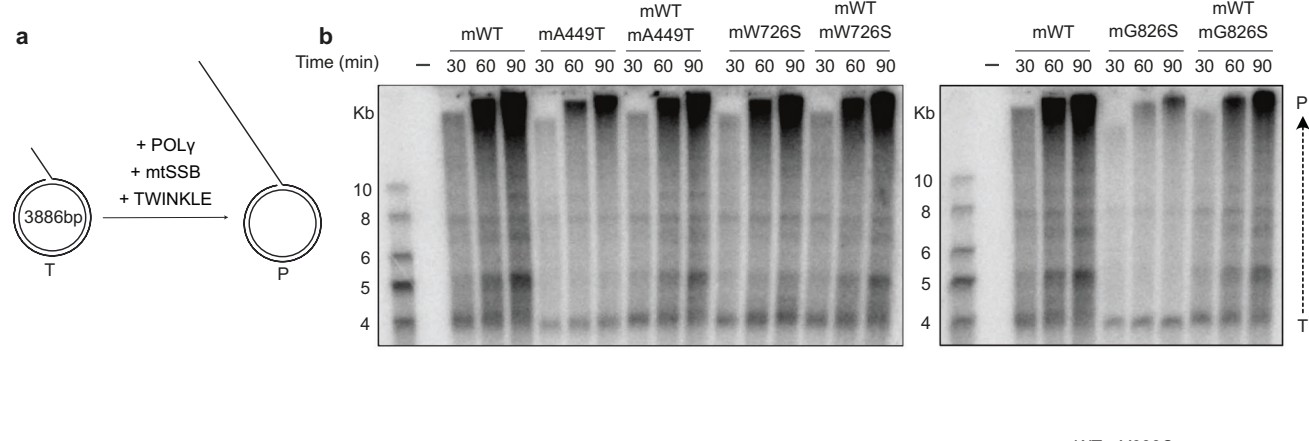

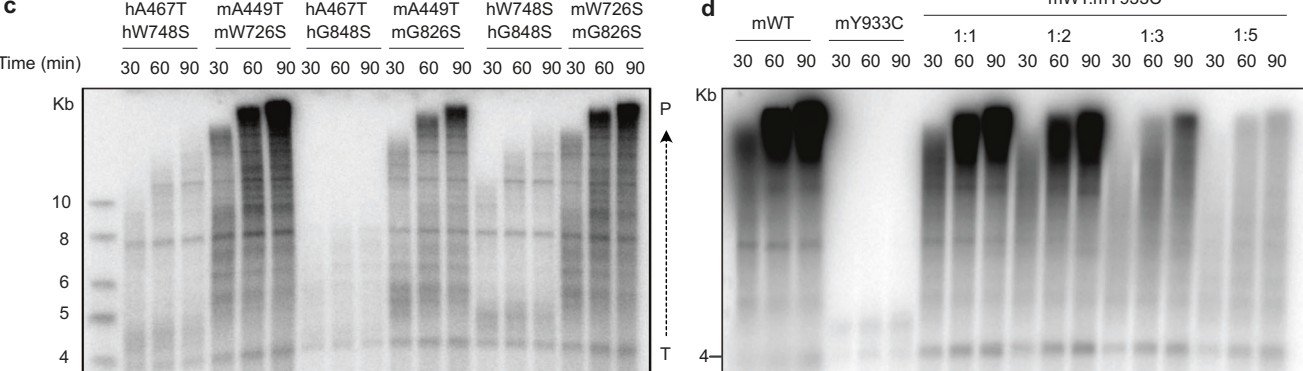

**Fig. 2 | The recessive mutations affect the mitochondrial mouse replisome.**
**a** Schematic representation of the rolling-circle assay used to investigate the effects of various mutant POLγ proteins on replisome function using a double-stranded DNA template. **b** Time-course experiments performed as outlined in **a** reveal that replication is compromised with all mutant forms of mPolγ variants when run in homozygous form during TWINKLE-dependent rolling-circle DNA replication. mPolγ$^{WT}$ or mutant mPolγ variants, as indicated, were incubated with TWINKLE, mtSSB, and the rolling-circle template at 37 °C for the indicated times. The polymerase concentration is always the same; when mixing two variants, half of each is used to match the final concentration in single-polymerase reactions. DNA synthesis was monitored as the incorporation of radioactive nucleotides, and the products of increasing length were separated on a 0.8% alkaline agarose gel. Template (T); Products (P). $n = 2$ independent experiments. **c** The experiments were performed as in (**b**), but the hPOLγ and mPolγ variants were combined as indicated. $n = 2$ independent experiments. **d** The time curve indicates that the mPolγ$^{Y933C}$ variant is unable to generate long stretches of DNA. The dominant effect of the mutant variant begins to impact mPolγ$^{WT}$ at a ratio of 1:1 but becomes more pronounced as the ratio of the mPolγ$^{Y933C}$ increases. $n = 2$ independent experiments. Source data are provided as a Source Data file.

synthesis, consistent with the recessive nature of these mutations (Fig. 2b and Supplementary Fig. 2a).

We next analyzed the effects on rolling-circle replication when recessive mutant variants were combined at a 1:1 ratio to mimic the compound heterozygous state often observed in affected patients (Fig. 2c). The heterozygous combinations included hPOLγ$^{A467T/W748S}$, hPOLγ$^{A467T/G848S}$, and hPOLγ$^{W748S/G848S}$, as well as the corresponding mouse combinations: mPolγ$^{A449T/W726S}$, mPolγ$^{A449T/G826S}$, and mPolγ$^{W726S/G826S}$. All human combinations exhibited more severe defects than their corresponding mouse counterparts, with hPOLγ$^{A467T/G848S}$ showing the most pronounced impairment, a trend also observed in the equivalent mouse combination (Fig. 2c).

Finally, we analyzed the impact of the dominant hPOLγ$^{Y955C}$ mutation on rolling-circle replication. As expected, the mutant polymerase on its own did not support DNA synthesis at physiologically relevant dNTP concentrations (Supplementary Fig. 2b). To investigate its dominant-negative effect, we maintained hPOLγ$^{WT}$ at a constant concentration, while adding increasing amounts of hPOLγ$^{Y955C}$. Interestingly, when hPOLγ$^{Y955C}$ was mixed with hPOLγ$^{WT}$ at a 1:1 ratio, mimicking the POLγ composition found in patients, DNA synthesis was only mildly reduced. However, consistent with its dominant-negative nature, increasing hPOLγ$^{Y955C}$ concentrations led to a gradual decrease in DNA synthesis. Similarly, mPolγ$^{Y933C}$ exhibited a loss of polymerase activity and reduced DNA synthesis when mixed with mPolγ$^{WT}$ (Fig. 2d).

This observation aligns with the fact that dominant mutations typically cause later onset and less severe phenotypes, with mtDNA deletions rather than depletion in affected patients[6]. As a comparison, we also tested the effect of adding hPOLγ$^{WT}$ or recessive mutants at a 5-fold excess over hPOLγ$^{WT}$ and found no significant impact, except for hPOLγ$^{G848S}$, which caused a mild reduction in DNA synthesis (Supplementary Fig. 2c). Similar results were obtained with the corresponding mouse variants (Supplementary Fig. 2d).

Overall, our data demonstrate that introducing human disease-causing mutations into mPolγ results in a similar molecular phenotype. However, in general, the effects are less severe with the mouse proteins compared to their human counterparts, highlighting both biochemical similarities and differences between the mouse and human mtDNA replication machineries.

## Structural differences between mPolγ$^{WT}$ and hPOLγ$^{WT}$

To better understand the differences between mouse and human POLγ, we employed single-particle cryogenic electron microscopy to determine the structure of the mPolγ$^{WT}$-DNA complex. Following the approach used in previous structural studies of hPOLγ$^{WT}$-DNA complexes, we introduced a single mismatch in the DNA substrate to induce POLγ to shift from replication to proofreading mode, capturing three conformations: the replication (R) conformer, the error-editing (E) conformer, and an intermediate (I) conformer[24].

As expected, mPolγ$^{WT}$-DNA complex adopted all three different conformations, with reconstructions refined to resolutions of 2.7–3.0 Å (Supplementary Figs. 3, 4a–c and Supplementary Table 1).

The R conformer was nearly identical between mouse and human, with a conserved catalytic polymerization site and structural architecture near the mutation sites (Fig. 3a–f and Supplementary Fig. 4d–g). The I conformer was also similar; however, the catalytic site and Y933 residue could not be modeled in the mouse structure due to high flexibility in the fingers subdomain (Supplementary Fig. 5a–g). Additionally, we observed a slight difference in the position of the distal POLγB subunit (Supplementary Fig. 5a).

When comparing the E conformers, we found that the overall structure of the POLγA subunit, including the exonuclease (EXO) site and mutation sites, was nearly identical in both species (Supplementary Fig. 6a–i). However, while the POLγA subunits remained largely unchanged in the E conformers, we identified a substantial shift in the positioning of the POLγB subunit between mouse and human (Supplementary Fig. 6a).

Our structural data indicate that the overall architecture of mPolγ$^{WT}$ and hPOLγ$^{WT}$ in replication mode is unchanged, and the catalytic subunit behaves similarly in both species during the transition from replication to error-editing. The four mutation sites are structurally well-preserved, and the mutations are therefore likely to have similar effects in both the mouse and human protein. However, in the E conformer—and to some extent in the I conformer—the accessory POLγB subunit occupies distinct positions relative to POLγA, highlighting a key structural difference between the human and mouse complexes.

## Differences in E conformers mouse and human POLγB subunits

Our findings prompted a closer analysis of POLγB subunits in the E conformers of human and mouse POLγ. Sequence comparisons of mPolγ and hPOLγ reveal that non-conserved residues cluster in POLγB and the L-helix of POLγA (Fig. 4a), a key part of the AID (accessory interacting determinant) subdomain responsible for POLγA–POLγB interactions[18]. When comparing the E conformers of the two species, the most significant positional shift occurs in POLγB and the L-helix, which move together in both structures (Fig. 4b). The comparison also reveals a minor displacement (2–4 Å) in the region interacting with the AID subdomain (Fig. 4c). However, from this region, a -10° rotation distinguishes mouse and human POLγB (Fig. 4d), leading to a more pronounced shift (8–10 Å) in areas distant from the AID contact surface. This shift is primarily observed in the distal POLγB monomer and the peripheral regions of the proximal monomer, with a greater displacement occurring in the human enzyme (Fig. 4e). Interestingly, the distal monomer has been shown to enhance the polymerization rate of POLγ due to additional POLγA interactions that stabilize the replication mode[24,25]. The observed differences in the positioning of the distal POLγB in the E conformers suggest that these structural variations may have functional consequences (see "Discussion").

Next, we investigated how the POLγB subunits and the AID subdomain move between the R and E conformers to determine whether internal movements occur in these regions during the transition from replication to error-editing mode (Fig. 4f, g). In mice, the POLγB subunits and the AID subdomain are nearly identical in the R and E conformers, indicating that these regions rotate as a rigid body relative to the rest of the POLγA subunit during the transition (Fig. 4f). This movement is driven by the rotation of the thumb helix subdomain, which causes the distal POLγB subunit to lose its interaction with POLγA, thereby promoting the E conformer[24]. The human complex behaves similarly; however, in addition to rigid body motion, there is also internal movement within the distal monomer (2–3 Å) and the AID subdomain (4 Å) (Fig. 4g). This suggests that these regions are more stable in mPolγ but more flexible in hPOLγ. Notably, the A467 residue is located near the AID subdomain, which shifts by 4 Å in the human

enzyme during the replication-to-error-editing transition (Fig. 4g). Since the phenotype of the mouse A449T mutation is relatively mild compared to the human A467T variant (Fig. 1g, h), it is possible that the rigidity of the mouse AID subdomain provides tolerance against the A449T variant.

To further explore species-specific differences, we assembled chimeric POLγ complexes using mouse POLγA with human POLγB (mAhB) and vice versa (hAmB), then analyzed their conformations by cryo-EM (Supplementary Figs. 7–10 and Supplementary Table 1). While R conformers resembled their wild-type counterparts, E conformers revealed key distinctions. The mAhB complex adopted both human-like and mouse-like E conformations, showing that human POLγB can induce a human-like conformation, but the accessory subunit alone does not dictate species differences (Supplementary Figs. 7 and 11). In contrast, the hAmB complex exclusively adopted the mouse-like E conformer, with some intermediates lacking DNA in the exonuclease site (Supplementary Fig. 9). The DNA density in this conformation was weaker than in mAhB, suggesting reduced stability. Overall, hPOLγA, when paired with mPOLγB, adopts a conformation more similar to the complete mouse enzyme (mAmB), whereas mPOLγA remains partially resistant to structural influence from hPOLγB during proofreading (Supplementary Fig. 11).

## mPolγB enhances the thermal stability of mPolγA

Since the accessory subunit plays a key role in POLγA stability[26] and structural differences exist between the human and mouse systems, we performed a thermofluor stability assay to assess WT and mutant PolγA in the presence or absence of PolγB. The mutants showed no major difference in melting temperature ($T_m$) compared to mPolγA$^{WT}$, but partial unfolding at lower temperatures was observed for mPolγA$^{A449T}$, mPolγA$^{W726S}$, and mPolγA$^{G848S}$ (Supplementary Fig. 12a–d). In complex with mPolγB$^{WT}$, mPolγA$^{WT}$ and mPolγA$^{Y933C}$ were strongly stabilized (-9 °C increase in $T_m$), while mPolγA$^{A449T}$, mPolγA$^{W726S}$, and mPolγA$^{G848S}$ showed a weaker stabilizing effect (-2–6 °C), suggesting reduced complex stability (Supplementary Fig. 12e–h).

Comparing the thermal stability of mouse and human POLγA$^{WT}$, POLγB$^{WT}$, and their respective complexes, we found that while the individual human subunits had higher $T_m$ values (44.5 °C for hPOLγA and 82.0 °C for hPOLγB, compared to 40.4 °C and 72.0 °C in mouse, Supplementary Fig. 12i, j, m, n), the mouse A–B complex was more stabilized upon binding of the accessory subunit (Supplementary Fig. 12k, o). Notably, switching accessory subunits in chimeric complexes did not affect $T_m$, indicating that the stability difference originates from POLγA rather than POLγB (Supplementary Fig. 12l, p).

The L-helix in the AID subdomain, which directly interacts with the accessory subunits, likely contributes to this stability difference (Supplementary Fig. 12q, r). Notably, T525 and R529 in mPolγA can form stabilizing electrostatic interactions with mPolγB, whereas their human counterparts (M544 and C548) lack this capacity (Supplementary Fig. 12s–u). These findings suggest that the AID–POLγB interaction is stronger in mice, leading to a more stable PolγA–PolγB complex, a result that correlates well with the thermofluor data. Moreover, the AID subdomain is a known LonP1 degradation target[26], and binding to the accessory subunit likely stabilizes the AID and protects it from degradation.

## mPolγB accounts for the increased activity in mPolγ

Our structural analysis and observed differences prompted us to investigate the molecular basis of the higher activity in mPolγ compared to hPOLγ. Despite significant structural variations in the error-editing mode, exonuclease activity was similar between human and mouse POLγ, and both enzymes displayed comparable DNA-binding affinity (Supplementary Fig. 13a–f). However, when measuring dNTP incorporation rates, mPolγ$^{WT}$ exhibited a significantly higher $k_{cat}$ than hPOLγ$^{WT}$ in its holoenzyme form (Fig. 4h, j). In contrast, when analyzing

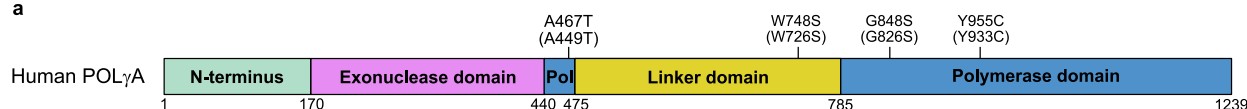

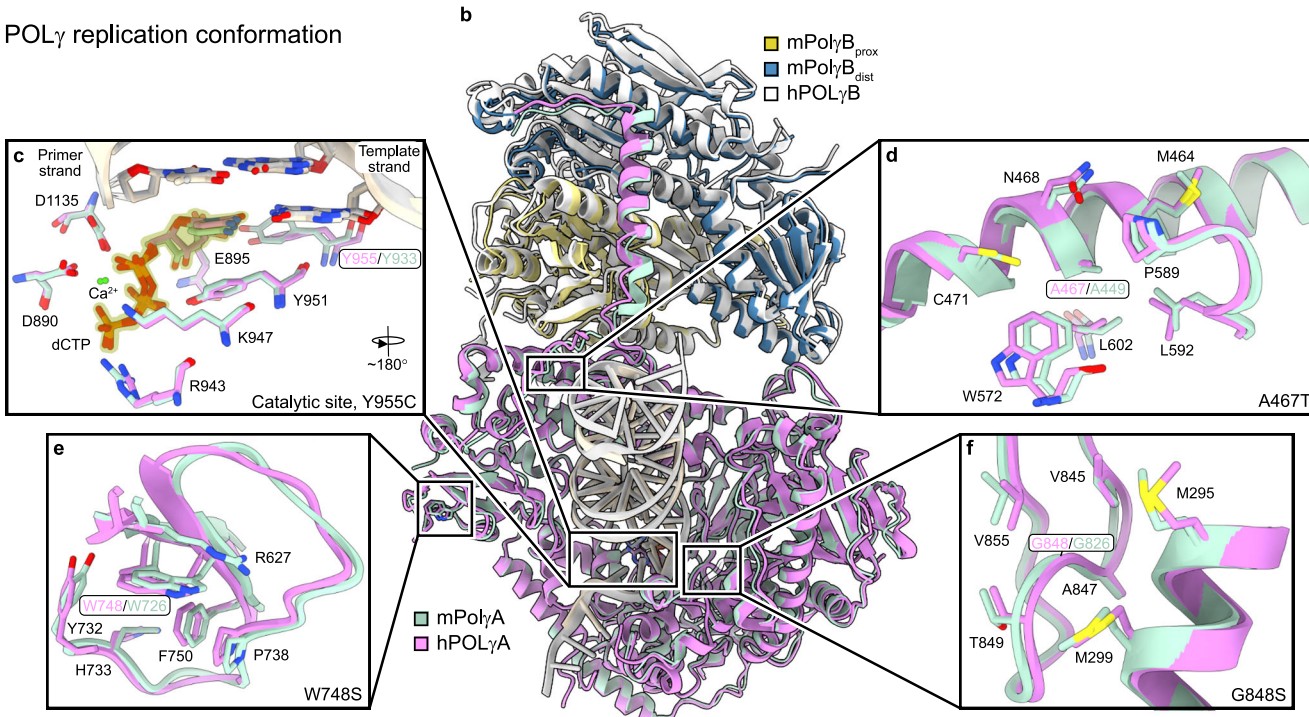

**Fig. 3 | Structural comparison of the hPOLγ and mPolγ replication conformers.** **a** Schematic overview of the domains in hPOLγA with the positions of the four disease variants marked, and the corresponding residues in mPolγA in parentheses. **b** Overview of overlaid hPOLγ (PDB ID: 8D37) and mPolγ replication conformers. The positions of the subunits; hPOLγA (magenta) and the hPOLγB dimer (white), are near identical to their mouse counterparts; mPolγA (green), proximal mPolγB (yellow) and distal mPolγB (blue). In both structures, the DNA is also in the same position. The DNA is colored gray (human) or beige (mouse). **c** Closeup of the catalytic site (POL) and the Y955/Y933 residues. The incoming nucleotide (dCTP) is highlighted in yellow. **d** Closeup of the A467/A449 position and surrounding residues. **e** Closeup of the W748/W726 position and surrounding residues. **f** Closeup of the G848/G826 position and surrounding residues. In **c**–**f**, the surrounding residues are numbered according to the human sequence.

hPOLγA$^{WT}$ and mPolγA$^{WT}$ alone, no significant differences in kinetic parameters were observed (Fig. 4i, j). Notably, kinetic measurements of the isolated POLγA subunit were performed without mtSSB, as mtSSB inhibits POLγA-mediated DNA synthesis (unpublished observation).

We hypothesized that mPolγB is a more potent accessory factor than its human counterpart. To test this, we created chimeric enzymes by swapping POLγB, forming hAmB (hPOLγA + mPolγB) and mAhB (mPolγA + hPOLγB). Strikingly, mPolγA paired with hPOLγB exhibited a reduced $k_{cat}$ compared to the complete mouse holoenzyme, while hPOLγA combined with mPolγB showed a modest but steady increase in activity (Fig. 4h, j). The enhanced DNA synthesis rate in mouse variants was further confirmed by comparing hPOLγ$^{G848S}$ and mPolγ$^{G826S}$ (Supplementary Fig. 13g, h). These findings suggest that mPolγB strongly enhances DNA synthesis, potentially mitigating the functional impact of disease-associated mutations in mPolγA.

Since processivity, the ability of the polymerase to remain attached to the DNA template during strand synthesis, could explain the difference, we analyzed processivity and DNA dissociation kinetics. However, no major differences between the polymerases were observed (Supplementary Fig. 14a). In fact, the mouse protein exhibited a slightly higher off-rate ($k_{off}$) than its human counterpart (Supplementary Fig. 14b).

To further assess the role of mPolγB in a replication context, we used a rolling-circle DNA synthesis assay. Consistent with our kinetic analysis, the wild-type mouse replisome was more active than the human counterpart (Supplementary Fig. 14c). Replacing hPOLγB with mPolγB increased rolling-circle replication, while the reverse swap reduced DNA synthesis. This pattern persisted when testing disease-causing variants of hPOLγA and mPolγA (Supplementary Fig. 14c, lower panels).

Clearly, mPolγB plays a key role in the enhanced activity of the mouse holoenzyme. However, mPolγA and its mutant variants also exhibit slightly higher activity than their human counterparts, regardless of whether they are paired with mPolγB or hPOLγB. The stronger stimulatory effect of mPolγB amplifies this difference, partially masking the negative impact of disease-causing mutations in the mouse system compared to the human system.

**Generation of mouse POLG models for common human variants**
Given our in vitro analysis, we set out to generate a collection of mouse models for these mutations to better understand the pathogenesis of *POLG*-related disorders. We used CRISPR/Cas9 to generate two knock-in mouse models harboring the *Polg*$^{G826S}$ and *Polg*$^{Y933C}$ mutations. We crossbred these mutants with two previously generated *Polg*$^{A449T}$ and *Polg*$^{W726S}$ mutations mice to obtain homozygous and compound heterozygous animals (Supplementary Fig. 15a, b)[11]. As most mutations are

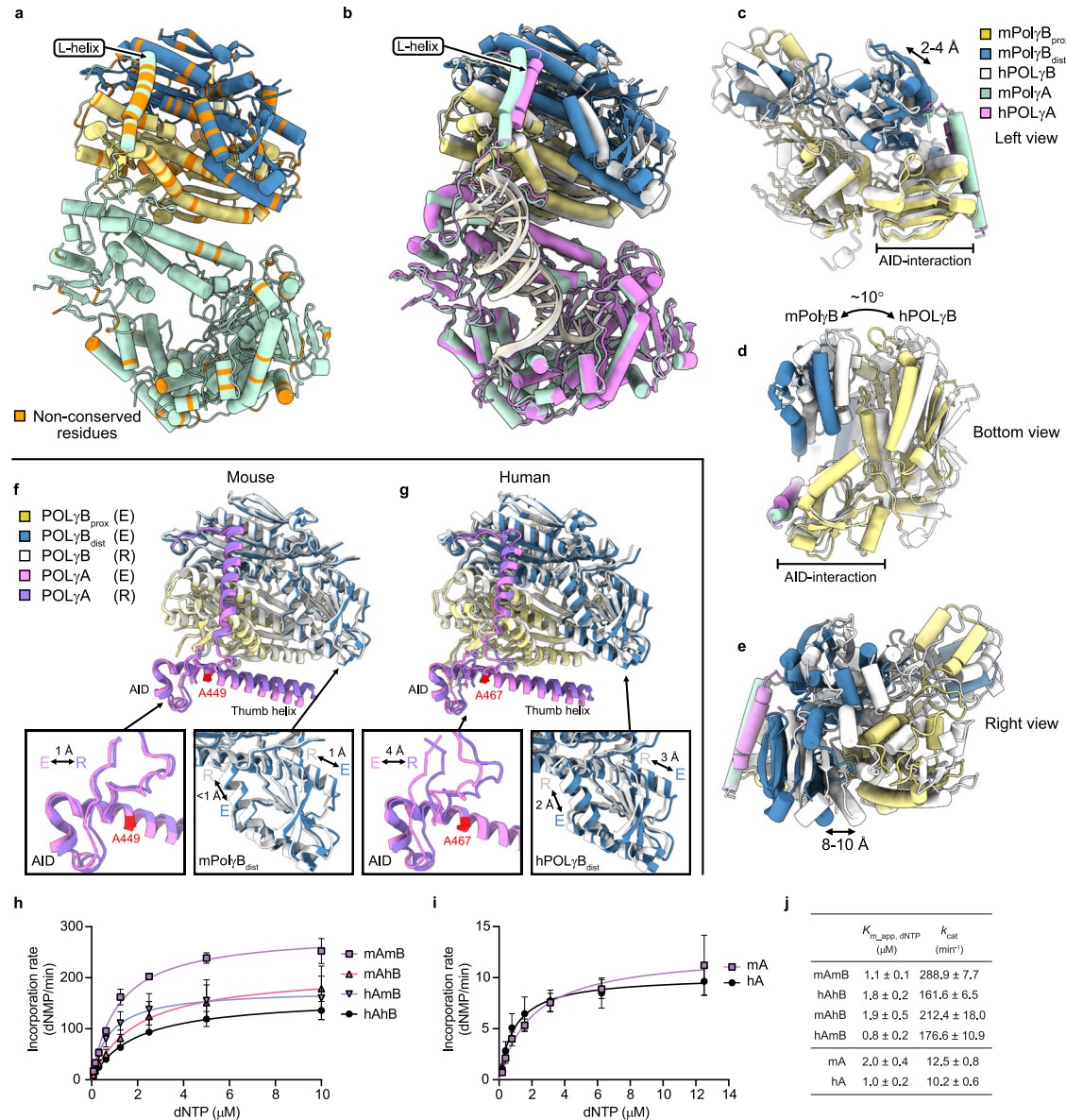

**Fig. 4 | The POLγB position differs in the human and mouse error-editing conformers. a** Overview of mPolγ (DNA omitted). Non-conserved residues between human and mouse POLγ are shown in orange. Most non-conserved residues are clustered in POLγB and the L-helix. **b** Overlaid hPOLγ (PDB: 8D42) and mPolγ error-editing conformers. The position of DNA, hPOLγA and mPolγA subunits are nearly identical in the two structures. However, the hPOLγB is shifted compared to both the distal and proximal mPolγB subunits. The L-helix in POLγA is also shifted as it interacts with POLγB. The DNA is colored gray (human) or beige (mouse). **c–e** Analysis of the shift between the POLγB subunits. The shift is less prominent in the proximal subunit region which interacts with the POLγA AID subdomain (**c, d**), with a small shift of 2–4 Å. However, peripheral regions, relative to the AID contact surface, experience a larger shift (8–10 Å) due to a rotation of ~10° (**d, e**). **f** Comparison of the POLγB subunits, the AID subdomain and the thumb helix in the R and E conformers in mouse. The region is stable and moves as a rigid body during the transition from replication to error-editing mode. **g** Comparison of the POLγB subunits, the AID subdomain and the thumb helix in the R and E conformers in human. A shift can be observed in the AID subdomain (4 Å) and the distal POLγB monomer (2–3 Å). The region is thus more flexible than its mouse counterpart during the transition from replication to error-editing mode. **h** Steady-state kinetics of dNTP incorporation was determined in the mouse system (mAmB), the human system (hAhB) and mixtures of human POLγA and mouse PolγB (hAmB), or mouse PolγA and human POLγB (mAhB) (mean ± s.d.; *n* = 3 technical replicates). **i** Steady-state kinetics of dNTP incorporation in human and mouse POLγA alone (mean ± s.d.; *n* = 3 technical replicates). **j** Kinetic parameters were obtained from the experiments in (**h**) and (**i**) (mean ± s.e.m.; *n* = 3 technical replicates). Source data are provided as a Source Data file.

strictly recessive in humans, we did not analyze the heterozygotes. However, we analyzed the *Polg*$^{+/Y933C}$ mice as the Y955C mutation is dominant in the patients.

These mutants can be grouped in three categories: (1) homozygous knock-in mice for the different mutations; (2) mutants crossed with *Polg*$^{+/KO}$ mice to generate compound heterozygotes with a null allele; (3) compound heterozygous mutants combining the different pathogenic mutations. We thus generated a unique collection of *Polg* genotypes covering the most common mutant combinations found in patients, accounting for about 80% of cases.

## Homozygous mutations in *Polg* mice

*Polg*$^{KO/KO}$ were previously reported as embryonic lethal[27]. *Polg*$^{A449T/A449T}$ and *Polg*$^{W726S/W726S}$ mice were born at Mendelian ratios (Supplementary Fig. 16a, b). In contrast, no *Polg*$^{Y933C/Y933C}$ and *Polg*$^{G826S/G826S}$ mice were obtained, indicating fully penetrant preweaning lethality (Supplementary Fig. 16c, d). This result is consistent with the dominant nature

of the Y955C mutation, which has never been observed in homozygosity or in association with other mutations (https://tools.niehs.nih.gov/polg/). Similarly, homozygosity for the G848S mutation has been reported in only four cases, typically resulting in death within a few days after birth[22]. These findings demonstrate that both Y933C and G826S mutations severely impair Polγ activity and are lethal in mice when present in homozygosity.

## Compound heterozygous mutations in *Polg* mice

To reproduce the genetic complexity of *POLG*-related disorders, we carried out an extensive breeding program to generate mice with combinations of the different alleles. We first crossed the *Polg*[+/A449T] with the *Polg*[+/KO] mice to generate the *Polg*[A449T/KO] genotype, a relatively frequent combination in the patients. The *Polg*[A449T/KO] mutants were born at the expected Mendelian frequencies (Supplementary Fig. 16e). We did not cross the other mutations with the KO allele as the corresponding combinations in humans are extremely rare or never found (https://tools.niehs.nih.gov/polg/).

We then crossed the *Polg*[+/A449T] mice with the *Polg*[+/Y933C] and, as expected, we found fully penetrant embryonic lethality (Supplementary Fig. 16f).

We generated the *Polg*[A449T/W726S], *Polg*[A449T/G826S] and *Polg*[W726S/G826S] genotypes, which are relatively frequent in patients presenting with extremely severe AHS or Leigh disease[21,28,29], by crossing the corresponding heterozygous mice. Whilst the *Polg*[A449T/W726S], *Polg*[A449T/G826S] mice were viable and born at the expected Mendelian frequencies, the *Polg*[W726S/G826S] were embryonic lethal (Supplementary Fig. 16g–i).

Notably, none of the viable homozygous and compound heterozygous models showed any obvious phenotype at birth, in contrast to the severity of the corresponding human syndromes. However, late onset phenotypes may arise, as shown below for the Y933C mutant.

In summary, we generated nine mouse models harboring various *Polg* mutations in different combinations, encompassing 70–80% of *POLG* patient genotypes. Consistent with the biochemical features and severity of patient symptoms, 4 out of the 9 *Polg* mutation models are embryonic lethal. Among these, the W748S/G848S combination contrasts with the human condition where patients survive but experience very early onset symptoms starting in infancy or childhood. The remaining five *Polg* models survive and provide a platform for investigating the pathogenesis of *POLG* disorders associated with different mutations.

## Effects of Polg mutations on mtDNA

We subsequently analyzed the amount and integrity of mtDNA in the tissues of the different *Polg* mice at 3 months of age. Real time PCR showed that mtDNA was mildly depleted in the skeletal muscle, but not in brain and liver of *Polg*[A449T/A449T] and *Polg*[A449T/KO] mice (Fig. 5a, b). *Polg*[W726S/W726S] mice had a 30% depletion in the liver but not in brain and skeletal muscle, as previously reported[10] (Fig. 5c). The *Polg*[+/Y933C], *Polg*[A449T/W726S] mutants had normal mtDNA content in all tissues (Fig. 5d, e). Importantly, the *Polg*[A449T/G826S] animals showed marked reduction of mtDNA copy number in all tissues (Fig. 5f).

Long-range (LR)-PCR was used to amplify the full length mtDNA from all tissues of each model. However, no multiple deletions were detected even after overexposing the gel (Supplementary Fig. 17).

Based on the mtDNA copy number, we investigated the effect of reduced mtDNA content on the OXPHOS activities in skeletal muscle homogenates from *Polg*[A449T/A449T] and *Polg*[A449T/KO] animals, in liver of *Polg*[W726S/W726S] mice, and in brain, liver and skeletal muscle of *Polg*[A449T/G826S] mice. Complex I and IV activities were normal in *Polg*[A449T/A449T] and reduced by 30–40% in the skeletal muscle of *Polg*[A449T/KO] mice. In addition, *Polg*[A449T/G826S] animals had decreased cIV activity in the liver but not in other tissues (Supplementary Fig. 18).

## Deep phenotypization of dominant Polg[+/Y933C] mutants

We investigated at what stage the *Polg*[Y933C/Y933C] mice died during embryonic development. We could detect homozygous individuals only at 7.5 d.p.c. (Supplementary Table 2).

We next carried out extensive in vivo phenotyping of the *Polg*[+/Y933C] mice, by using the pipeline shown in Supplementary Fig. 19a. We used indirect calorimetry and measured respiratory gases, heat production and respiratory exchange ratio by whole body calorimetry but did not detect any difference between wild-type and mutant mice (Supplementary Fig. 19b). In addition, spontaneous movements were similar between WT and *Polg*[+/Y933C] mice (Supplementary Fig. 19b). We further confirmed that climbing and ambulatory activities were normal in *Polg*[+/Y933C] mice by using home cage analysis (HCA) (Supplementary Fig. 20a, b). The *Polg*[+/Y933C] mice showed normal rotarod performance, blood lactate and glucose levels, lean and fat mass, and body weight (Supplementary Fig. 21a–f). The levels of protein of the mitochondrial replisome were similar in WT and *Polg*[+/Y933C] tissues (Supplementary Fig. 22a). At this age, we did not observe gross abnormality in any tissue (Supplementary Fig. 22b).

We then performed indirect calorimetry and HCA in 16-month-old in *Polg*[+/Y933C] and wild-type littermates, but did not find abnormalities in VO2, VCO2, RER, heat production and spontaneous movements (Supplementary Figs. 23 and 24a, b). However, hematoxylin and eosin (H&E) analysis at 24 months of age revealed extensive vacuolization of the white matter in the cerebellum and brainstem of the mutant mice (Fig. 6a). Immunohistochemistry (IHC) using anti-CD68 and anti-GFAP antibodies revealed the presence of numerous activated microglial cells and extensive gliosis in the same areas showing vacuolization (Fig. 6a). These findings indicate ongoing neurodegeneration in 24-month-old *Polg*[+/Y933C] mice.

In addition, COX/SDH staining revealed the presence of mitochondria depleted areas in several *Polg*[+/Y933C] muscle fibers (Fig. 6b). H&E of liver sections at 24 months of age showed increased steatosis and inflammatory cells around the vessels in *Polg*[+/Y933C] but not wild-type littermates (Fig. 6b).

In contrast to a previous report showing that heart-specific overexpression of human Y955C mutant led to cardiomyopathy[30], we did not observe significant alternations in heart and kidneys at 24 months of age.

These findings are compatible with late onset phenotypes observed in patients harboring the Y955C mutation.

Since multiple mtDNA deletions were observed in Y955C patients, we analyzed mtDNA amount and integrity in the tissues of 24-month-old *Polg*[+/Y933C] mice. Both mtDNA integrity and copy number were normal (Fig. 7a, b). However, we observed clear ultrastructural alterations in mitochondria from skeletal muscle of *Polg*[+/Y933C] mice, including increased area, length and width, compared to wild-type littermates. Overall, we found a massively increased number of fibers showing aberrant mitochondria (Fig. 7c, d).

We finally analyzed the protein levels of the components of the mitochondrial replisome by western blot. The level of PolγA, PolγB, and Tfam, were similar in *Polg*[+/Y933C] and wild-type samples (Supplementary Fig. 24c).

## Discussion

To elucidate the molecular and phenotypic variability of POLG-related syndromes, we conducted a comprehensive in vitro and in vivo analysis of the most common disease-associated mutations in *POLG*. Our results establish a strong parallelism between human and mouse POLγ variants, underscoring the relevance of the mouse model for studying *POLG*-related diseases.

Enzymatic activity assays demonstrated that mutations followed a clear hierarchy of severity in both species: W748S < G848S < A467T < Y955C in humans and W726S < A449T < G826S < Y933C in mice. While the impact of mutations was consistently milder in the mouse system,

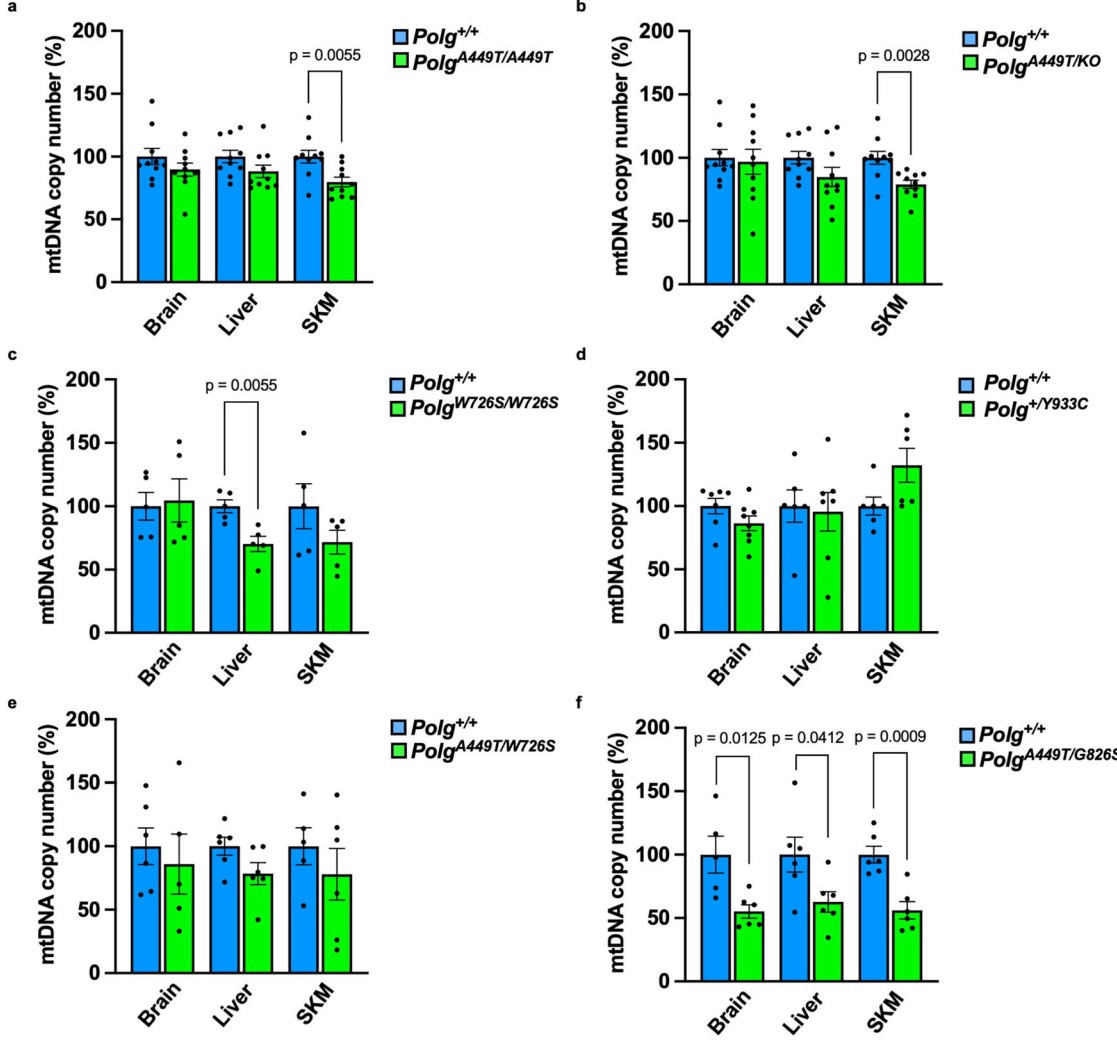

**Fig. 5 | mtDNA content in 3-month-old mouse tissues.** MtDNA content was analyzed in 3-month-old mice by real-time PCR. **a** *Polg^A449T/A449T* (*n* = 10/genotype), **b** *Polg^A449T/KO* (*n* = 10/genotype), **c** *Polg^W726S/W726S* (*n* = 5/genotype), **d** *Polg^+/Y933C* (*n* = 8/genotype), **e** *Polg^A449T/W726S* (*n* = 6/genotype), **f** *Polg^A449T/G826S* (*n* = 6/genotype)

mice. Both males and females were used. Data are presented as mean ± S.E.M. *p* values were calculated by two-tailed unpaired Student's *t* test. Outliers were identified by using Rout test and excluded from the analysis. Source data are provided as a Source Data file.

the overall trend remained conserved, emphasizing the fundamental similarities between the two species. Notably, the A449T mutation was relatively well tolerated in the mouse system compared to its human counterpart.

Combinations of these mutations led to a highly nuanced picture of increasing severity, which for the mouse mutants is: A449T/ W726S < W726S/G826S < A449T/G826S. In our in vivo data, the A449T/W726S mutants were born and had no clinical or molecular phenotype; the A449T/G826S mice were viable and showed partial depletion of mtDNA in all tissues, and the W726S/G826S mice were embryonic lethal. These findings align well with the corresponding human mutations, where patients with compound heterozygous A467T/G848S or W748S/G848S mutations typically exhibit severe early-onset disease. Interestingly, in vitro data suggested that A449T/ G826S had a more severe enzymatic defect than W726S/G826S, yet the latter exhibited greater embryonic lethality in vivo. This discrepancy suggests that factors such as POLγA protein stability and expression levels may contribute to disease severity beyond direct enzymatic impairment.

Furthermore, while the dominant Y933C mutant interferes with WT protein function, the recessive mutants do not exert this effect. Interestingly, the G848S mutant exhibits a slight negatively dominant

effect in vitro, raising the question of whether carriers of this mutation might develop a very mild mitochondrial phenotype later in life.

Our findings also reveal that all mutant combinations we examined developed clinically relevant phenotypes, either preventing normal embryonic development or causing early postnatal lethality. For the Y933C mutant, we detected homozygous embryos at 7.5 but not at 9.5 days post-coitum, indicating very early embryonic lethality. This observation mirrors findings from knockout mouse models for components of the replication machinery, such as *Twnk^-/-* and *Polg^-/-*, highlighting the critical role of mtDNA replication at this stage of development.

Importantly, mouse models exhibit in general milder phenotypes compared to human syndromes. This discrepancy can be attributed to intrinsic biochemical and structural differences between human and mouse enzymes. Both wild-type and mutant versions of mPolγ exhibit higher activity than their human counterparts, with the increased efficiency primarily driven by the mPolγB subunit. Notably, replacing mPolγB with its human counterpart significantly reduces enzyme activity, underscoring the importance of this accessory subunit in polymerase function.

Our cryo-EM studies reveal structural differences between the human and mouse enzymes in the error-editing state. In mice, the

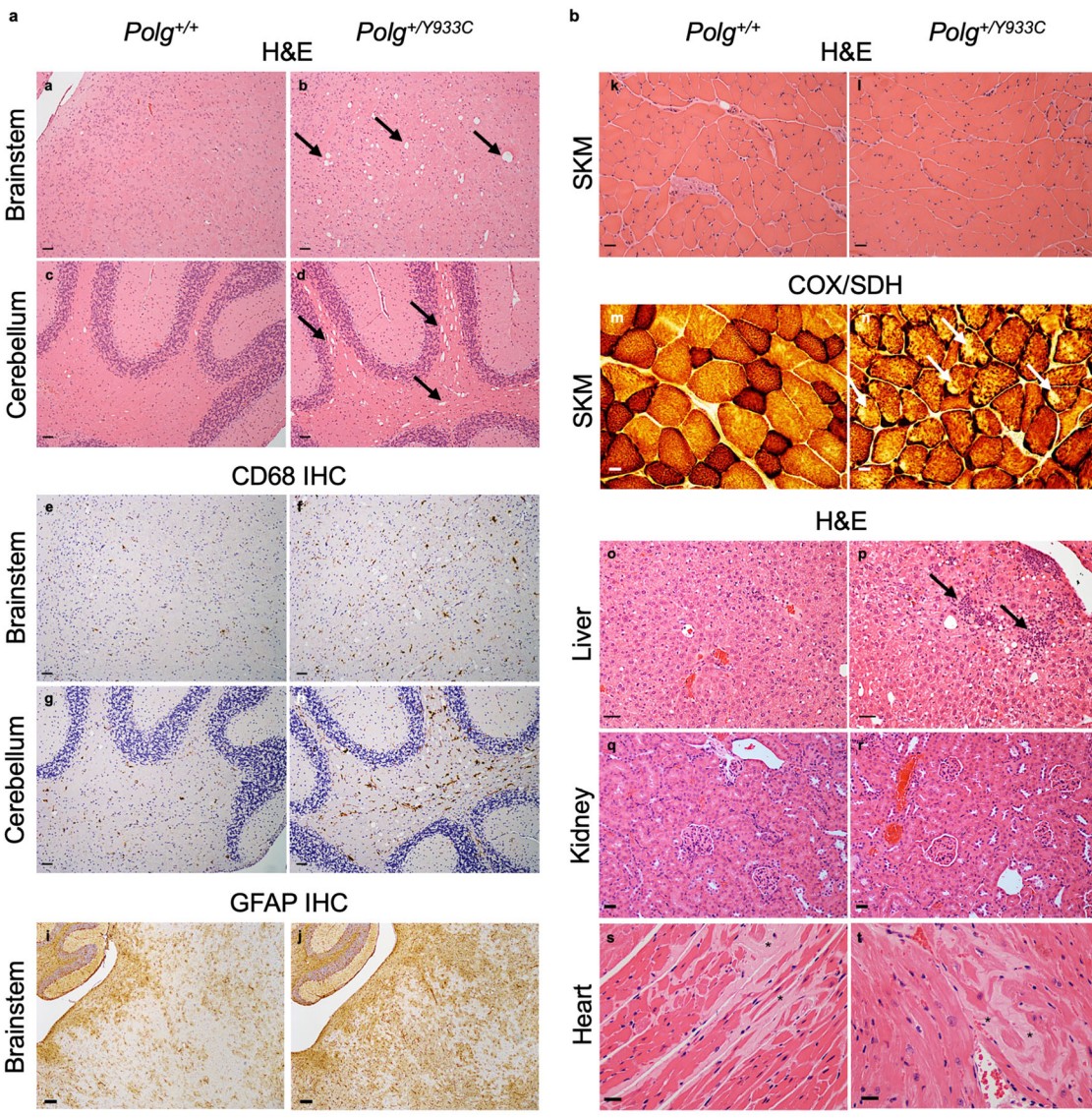

**Fig. 6 | Histological analysis of 24-month-old *Polg*^+/+ and *Polg*^+/Y933C mice. a** *a–d*: Representative H&E staining of brain sections from the brainstem and cerebellum. Note the presence of extensive vacuolization (arrows) in the brainstem and in the cerebellar white matter of heterozygous mice which is hardly visible in the control littermates. Scale bars, 50 μm. *e–h*: Representative CD68 immunohistochemistry of brain sections from brainstem and cerebellum showing the presence of numerous activated microglial cells in the brainstem and the cerebellar white matter of *Polg*^+/Y933C mice (colored in brown). Note that some positive cells are also present in the corresponding brain regions of WT littermates. Scale bars, 50 μm. *i,j*: Representative GFAP immunohistochemistry. Note the increased staining for GFAP in the brainstem of the heterozygous mouse. (brownish color). Scale bars, 100 μm. **b** *k, l*: representative H&E staining of skeletal muscle samples. No obvious alterations were present. *m, n*: Representative picture of double COX/SDH staining of skeletal muscle sections. Note the presence of mitochondria-depleted fibers in the heterozygous animals (white arrows). The absence of blue color indicates a general reduction of mitochondrial content. Scale bars, 20 μm. *o, p:* H&E pictures of liver sections. Note the increased steatosis and presence of inflammatory cells in the liver of the *Polg*^+/Y933C mouse (arrows). Scale bars: 40 μm. *q, r*: H&E pictures of kidney sections. No obvious alterations were present. Scale bars: 40 μm. *s, t*: H&E pictures heart sections. The asterisks highlight the presence of amyloidotic deposits in both *Polg*^+/+ and *Polg*^+/Y933C mice. Scale bars: 20 μm. Four animals of each genotype have been analyzed.

distal POLγB subunit and the AID subdomain exhibit less internal motion during the transition from replication to error-editing mode, which provides a structural basis for the milder phenotypic effects observed in mouse models. Consistent with this, Yin and colleagues have shown that the distinct functions of human POLγB are attributed to its dimeric structure, where the monomer closest to POLγA enhances DNA interaction, while the distal monomer increases the reaction rate[25]. Our findings extend this model by demonstrating that species-specific conformational differences influence polymerase efficiency and disease mutation tolerance.

This structural rigidity may also explain why the A449T mutation near the AID domain leads to a milder phenotype in mice compared to humans. Additionally, the reduced flexibility of the mouse holoenzyme appears to enhance nucleotide incorporation efficiency, as POLγB plays a critical role in stabilizing POLγA, improving DNA-template engagement, and accelerating catalytic turnover. Indeed, our kinetic measurements (Fig. 4h–j) show that even subtle repositioning of POLγB can significantly enhance polymerase activity, a mechanism that could broadly counteract the functional deficits caused by diverse POLγA mutations. Given the clinical heterogeneity of pathogenic POLγA variants, targeting POLγB could be a promising therapeutic strategy for restoring polymerase function across multiple disease-associated mutations.

Compensatory mechanisms may further contribute to the milder phenotypes observed in mice. For example, in *Mpv17* knockout mice, which display ~95% mtDNA depletion in the liver, mitochondrial

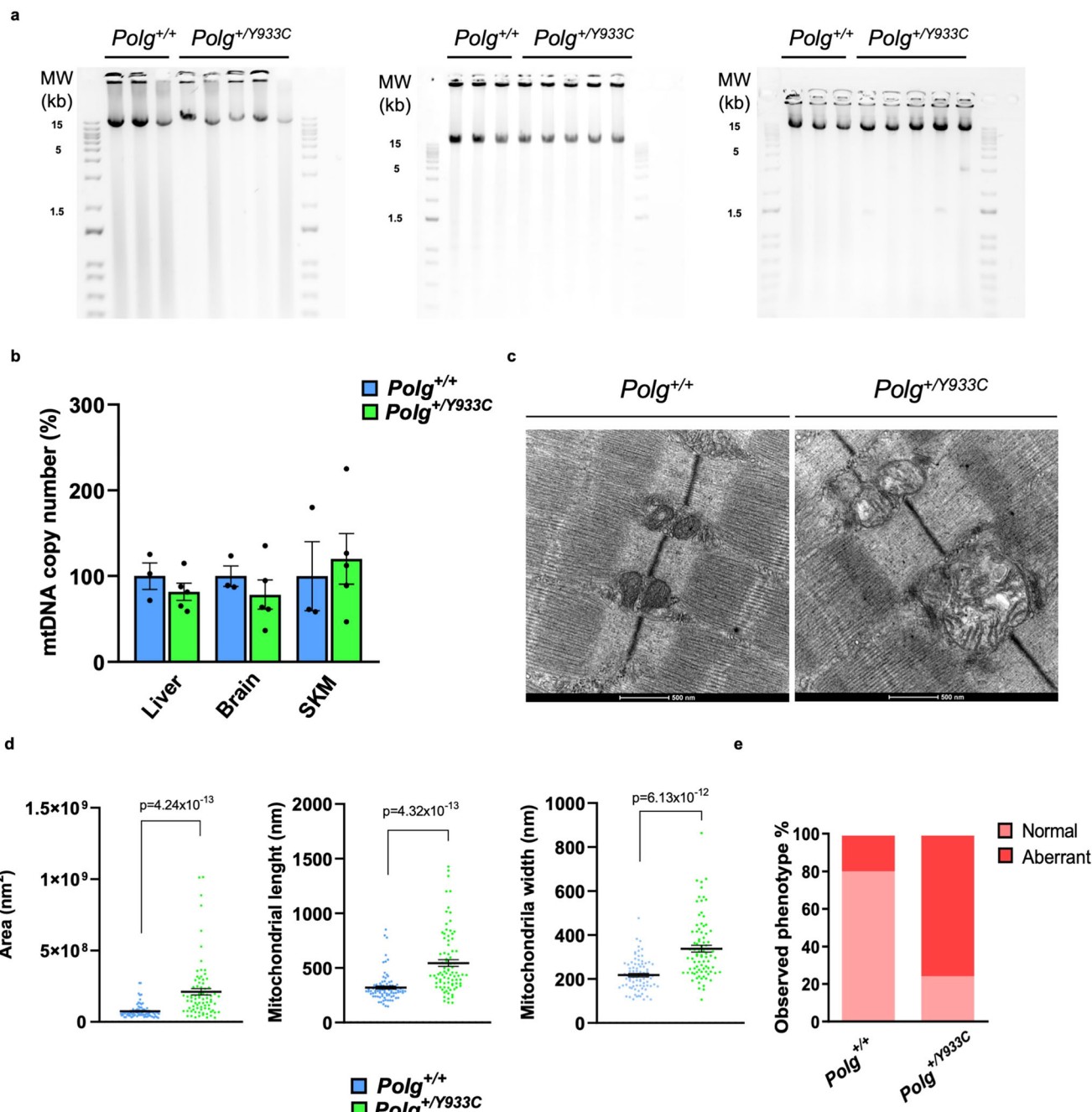

**Fig. 7 | Mitochondria and mtDNA analysis of tissues from 24-month-old mice. a** Long-range PCR performed in DNA isolated from brain, liver, and SKM of 24-month-old *Polg*[+/+] (*n* = 3) and *Polg*[+/Y933C] (*n* = 5) mice. Primers amplified a fragment of 15,781 bp of the mtDNA. The bands were visualized by SYBR™ safe staining. Each lane represents a biological replicate. Both females and males were used. **b** Real-Time qPCR quantification of mtDNA content in liver, brain, and SKM of 24-month-old *Polg*[+/+] (*n* = 3) and *Polg*[+/Y933C] (*n* = 5) mice. Data are presented as mean ± S.E.M. Both females and males were used. **c** Representative EM images from *Polg*[+/+] and *Polg*[+/Y933C] SKM. The size bar represents 500 nm. **d** Quantification of the mitochondrial area, length and width (*n* = 80 mitochondria/genotype, *n* = 15 independent TEM images per condition from 2 independent animals). Error bars indicate ±S.E.M. *p* values were calculated by two tailed unpaired Student's *t* test. **e** Quantification of mitochondria with aberrant morphologies in SKM. *n* = 15 independent TEM images per condition from 2 independent animals. Two-tailed $\chi^2$ test was used for statistical analysis. Source data are provided as a Source Data file.

transcripts remain relatively stable, being reduced by only 50% compared to wild-type littermates[31]. This highlights how compensatory pathways can help mitigate mtDNA depletion, suggesting that maintaining mtDNA levels above a critical threshold is crucial for preventing disease.

We conducted a comprehensive phenotypic characterization of *Polg*[+/Y933C] heterozygous mice. The human equivalent mutation Y955C, is associated with autosomal dominant progressive external ophthalmoplegia and muscle weakness, with muscle biopsies showing accumulation of ragged red fibers and multiple mtDNA deletions[32]. Additional symptoms may include Parkinsonism and ovarian failure. A previous mouse model overexpressed the human *Polg*[+/Y933C] in heart, leading to reduced mtDNA copy numbers and cardiomyopathy[30]. In our study, we did not notice cardiomyopathy. Our in vitro experiments showed that adding an excess of Y933C protein exacerbates the molecular phenotype (Fig. 2d), which is in agreement with and explains

why the overexpression of Y933C results in more severe phenotypes. In contrast, we detected late-onset neurological and neuromuscular phenotypes, including brain vacuolization, astrogliosis, activated microglia, and skeletal muscle fibrosis with altered COX staining. It is worth noting that we did not detect mtDNA deletions, which is a common observation in other mouse models of the mitochondrial replisome. For instance, the 'deletor' mouse, carrying a pathological dominant mutation in the mitochondrial helicase Twinkle associated with multiple deletions in patients, shows very low levels of deletions only from 20 months of age[33]. Similarly, the thymidine phosphorylase knockout mouse, associated with mitochondrial neurogastrointestinal encephalomyopathy (MNGIE) disease characterized by depletion and multiple deletions, does not accumulate mtDNA deletions[34].

The absence of deletions in these models may be due to the shorter lifespan of mice compared to humans, limiting the time available for deletions to accumulate. In support of this, the Y955C mutation causes a relatively late-onset disease in patients, suggesting that the defect needs to accumulate over time. Overall, our findings indicate that the shorter lifespan of mice prevents the development of the fully blown phenotype.

In summary, our work establishes mouse models as valuable tools for dissecting POLG-related disease mechanisms while revealing critical species-specific differences in polymerase structure and compensatory pathways that inform therapeutic strategies to mitigate mitochondrial dysfunction across mutations

## Methods
### Animal work
All animal experiments were carried out in accordance with EU Directive 2010/63/EU (authorization from the Italian Ministry of Health: PR474/2021). The animals were maintained in a temperature- and humidity-controlled animal care facility with a 12-h light/12-h dark cycle and free access to water and food, and they were monitored weekly to examine body condition, weight and general health. The mice were sacrificed by cervical dislocation at 3, 12 and 24 months of age for subsequent analysis. Number and sex of the animals used in the experiments is detailed in the legends.

The mice carrying the A449T mutation were on a FVB/J background, as previously reported. When animals from this strain were crossed with other mutants, which were on a C57Bl/6J background, F1 mixed FVBJ/C57BL6J mice were generated, and wild-type littermates used as controls. Ten to fifteen males and females were used for in vivo analysis; 4–10 mice per group were used in each experiment on molecular parameters as detailed in the legends. Males and females were analyzed separately for the in vivo tests where sex-related differences were present and were pooled for the in vitro analyzes.

### Generation of *Polg* mutant mice
*Polg*$^{A449T/A449T}$ mice were generated as described[11]. W726S mice were generated by homologous recombination as described in ref. [10]. The other recombinant animals were obtained by CRISPR/Cas9 technology. All gRNAs and donor oligonucleotides used in this study are listed in Supplementary Table 3. *Polg*$^{Y933C}$ mutants were generated by electroporation of gRNAs, donor oligo and Cas9 vector, in mouse ES cells as no mutant was obtained by direct injection of the reagents in the 1-cell embryos. ES cells screening identified seven recombinant clones, two of which were injected in the blastocysts of C57Bl/6 mice, then re-implanted in pseudo-pregnant females. Chimeric pups were screened by PCR and restriction digestion using the MscI restriction enzyme whose target sequence was on purpose included in the design of the donor oligonucleotide. Positive pups were backcrossed to C57Bl/6 wild-type mice to assess germ-line transmission.

The G826S mutants were generated by injection of the CRISPR/Cas reagents (Supplementary Table 3) in the blastocysts of C57Bl/6 mice, then re-implanted in pseudo-pregnant females. Recombinant

animals carrying the mutations were identified by PCR and restriction digestion with NheI, which was incorporated as a silent mutation on the donor oligonucleotide. Two positive individuals per strain were used to assess germ-line transmission as described above. Genotyping was routinely done by using the oligos listed in Supplementary Table 3. The expected bands are shown in Supplementary Fig. 15b. The PCR is carried using GoTaq DNA polymerase (Promega, UK) according to manufacturer instructions. Annealing temperature were as follow: *Polg*$^{A449T}$, 63.7 °C; *Polg*$^{W726S}$ and *Polg*$^{G826S}$, 60 °C; *Polg*$^{Y933C}$: 61 °C; *Polg*$^{KO}$: 55 °C.

### In vivo analysis
**Rotarod.** A rotarod apparatus (Ugo Basile) was used to assess coordination skills. During the test, mice had to maintain themselves on a rod turning at accelerating speeds. The latency to fall was recorded. The mice underwent three trial sections at least 20 min apart, using an acceleration protocol from 5 to 40 rpm in 300 s.

**Glucose measurement.** Mice were fasted for overnight and EMLA cream was applied to the base of the tail prior to sampling. A small incision was made in the lateral tail vein and a calibrated Alphatrak was used to measure blood glucose level in one drop of blood.

**Lactate measurement.** EMLA cream was applied to the base of the tail prior to sampling. A small incision was made in the lateral tail vein, and a calibrated Lactate Scout 4 analyzer was used to measure blood lactate level in one drop of blood before and after a main tail bleed was taken.

**DEXA scan.** The body composition of the mice was assessed using the Faxitron Ultrafocus machine (GE Medical Systems, USA). Mice were anaesthetized with ketamine/xylazine or with isoflurane. High energy X-Ray images were automatically analyzed for fat tissue content, lean tissue content, bone mineral density and bone mineral content. If mice were anaesthetized with ketamine and xylazine, the mice also underwent the auditory brain stem response test. More details available at IMPRESS.

**Indirect calorimetry.** The metabolic rate of the mice was assessed using indirect calorimetry. Mice were individually housed overnight for a period of 21 h in Phenomaster cages (TSE Systems, Germany) with standard bedding and red Perspex igloos. The air content in each cage was sampled for 1 min, in turn and a reference cage with no mice in it was sampled for comparison. The oxygen consumption and carbon dioxide production for each mouse was then determined as the difference between the reference amount of gas and the level in the mouse cage. Ambulatory activity of mice is also recorded using beam breaks.

VO2 and VCO2 are calculated based on the flow rate of air (liters per minute), the percentage change in gas mixture (dO2 or dCO2) and accounts for the temperature in each chamber as well. VO2, VCO2 and Heat are normalized to the body weight. Activity of the mouse is determined by beam break. For the activity values, 'A' represents ambulatory movement and 'F' represents fine movement. These are determined by the program depending on how the mouse breaks consecutive beams. The directions are given in the $X$ and $Y$ axis with the $Z$ axis equating to rearing. The activity is also split into zones−Central (Cen) and Periphery (Per). Activity measurements are continuous for the time bin and are not measurements per minute.

**HCA.** Radio frequency identification microchips were injected subcutaneously into the lower left or right quadrant of the abdomen of each mouse. These microchips were contained in standard ISO (FDXB) biocompatible glass capsules (12 × 2 mm; AVID Plc). The procedure was performed on sedated mice (Isoflo; Abbott, UK), butorphanol

opioid analgesic (Torbugesic; 0.025 mg/ml) was injected subcutaneously in the scruff after sedation but prior to microchip insertion and the wound sealed with tissue adhesive, GLUture. The animals were allowed to recover from the microchip procedure for at least 1 week before being placed in the HCA rigs for data collection. Three days prior to recording sessions, the animals were transferred to clean home cages with fresh bedding, nesting material and a cardboard rodent tunnel as enrichment material, in line with the standard husbandry procedures for IVC cages. The cages were then placed in an IVC rack in the experimental room for the animals to acclimatize. For each recording, the cages were randomly assigned to an HCA rig. Data were collected and analyzed as previously described[35].

**Gait analysis.** The assessment was performed using the DigiGait imaging apparatus (DigiGait, Mouse Specifics, Inc., Boston, MA). Mice were placed on a motor driven treadmill within a plexiglass compartment (~25 cm long and ~5 cm wide). A high-speed digital video camera was mounted underneath the transparent treadmill belt to visualize paw contacts. The treadmill speed was maintained at a constant 20 cm/s, a speed at which the animals were capable of running continuously. The resulting video footage (approximately 5 s in duration) was analyzed using DigiGait software to extract various gait parameters.

### DNA extraction and analysis

Genomic DNA was extracted by digesting the tissues in lysis buffer (0.5% sodium dodecyl sulfate (SDS); 0.1 M NaCl; 50 mM Tris–HCl, pH = 8; 2.5 mM EDTA) containing Proteinase K (20 ng/μL) overnight at 55 °C. The following day, RNase A was added to the mixture (100 μg/ml) and samples were incubated for 30 min at 37 °C. An isovolume of phenol:chloroform was added to the lysates, mixed gently, and samples were centrifuged at 4 °C for 15 min at $14{,}000 \times g$. Supernatant was collected into a new Eppendorf tube and mixed with 1/10 volumes of 3 M Sodium Acetate and 2 volumes of 100% EtOH. Samples were centrifuged at 4 °C for 15 min at $14{,}000 \times g$ and pellets were washed with 70% EtOH. Final DNA was resuspended in water.

For mtDNA relative quantification, SYBR Green real-time qPCR was performed using primers specific to a mouse mtDNA region in the COI gene. Primers specific to RNaseP, a single copy gene taken as a nuclear gene reference. All primers are listed in Supplementary Table S3. Approximately 25 ng of DNA was used per reaction.

For Long-range PCR, mtDNA was amplified from 50 ng of total DNA with the primers (LongR_mtDNA_Fw and LongR_mtDNA_Rv, Supplementary Table S3) using PrimeSTAR GXL DNA polymerase (TAKARA, Japan) and following PCR conditions: 98 °C for 10 s, 68 °C for 13 min, 35 cycles.

### Biochemical analysis of OXPHOS activities

Tissues stored in liquid nitrogen were homogenized in 10 mM of potassium phosphate buffer (pH = 7.4). The spectrophotometric activity of respiratory chain complexes I, and IV, as well as citrate synthase, was measured as follows[36]. Complex I activity was measured adding 1–5 μl of tissue homogenate in a buffer containing 250 mM sucrose, 2 mM EDTA, 100 mM Tris–HCl pH 7.4, between 10 and 100 μM decyl-ubiquinone and 2 mM KCN at 30 °C. Reactions were started by the addition of 50 μM NADH and oxidation followed at 340 nm for 2 min. Rotenone was used to subtract any rotenone insensitive activity.

Complex IV activity was measured by adding 1–5 μl of homogenate to a mix containing 10 mM $KH_2PO_4$, pH 7.2 and between 5 and 50 μM ferrocytochrome c. The rate of oxidation of ferrocytochrome c was recorded for 2 min (extinction coefficient of 27.2 mM$^{-1}$ cm$^{-1}$).

Citrate synthase activity was measured at 30 °C by using 1–5 μl of homogenate in a reaction mixture containing 125 mM Tris–HCl, 100 μM DTNB (5,5'-dithiobis(2-nitrobenzoic acid) and 300 μM acetyl coenzyme A. The reaction is initiated by the addition of 500 μM oxaloacetate, and DTNB reduction at 412 nm measured for 2 min. The

mitochondrial respiratory chain activities were expressed as nmoles/min/mg of protein.

### Histological analysis

Mouse tissues for Haematoxylin and Eosin (H&E) analysis, were fixed in 10% neutral buffered formalin (NBF) for a few days at room temperature and then included in paraffin wax. Sections of 4 μm were used for analysis. H&E staining was performed by standard methods.

For COX/SDH histochemical analysis, SKM (gastrocnemius) samples were frozen in isopentane pre-cooled in liquid nitrogen. Sections of 8 μm were stained for COX and SDH activity[37]. For SDH activity, 8 μm thick sections from frozen tissue were collected on coverslips. The reaction mix contained: 5 mM phosphate buffer pH 7.4, 5 mM EDTA, 1 mM KCN, 0.2 mM Phenazine methosulfate (PMS), 50 mM Succinic acid, 1.5 mM Nitro blue tetrazolium (NBT). NBT was the electron acceptor with PMS served as intermediate electron donor to NBT. Sections were incubated for 20 min at 37 °C.

For Cytochrome-c Oxidase the sections were incubated 1 h at 37 °C. The reaction mix contained: 5 mM phosphate buffer pH 7.4, 0.1% 3,3'-Diaminobenzidine (DAB), 0.1% Cytochrome c, 0.02% catalase.

### Western blot immunovisualization and antibodies

Mouse tissues were homogenized in RIPA buffer [150 mM sodium chloride, 1.0% NP-40, 0.5% sodium deoxycholate, 0.1% SDS, 50 mM Tris, pH 8.0] in the presence of protease inhibitors (cOmplete™ Protease Inhibitor Cocktail, Sigma). Protein concentration was determined by the Lowry method. Aliquots, 30 μg each, were run through a NuPAGE™ 4–12% Bis-Tris gel (Invitrogen, NP0321BOX) and electroblotted onto a polyvinylidene fluoride (PVDF) membrane, which was then immunodecorated with different primary antibodies: anti-POLγA (1:500) was from Cell Signaling (D1Y6R), anti-POLγB (1:500) was from Novus (NBP2-94064), anti-GAPDH (1:5000) was from GeneTex (GTX627408), anti-TFAM (1:2000) was from Abcam (ab131607). Secondary antibodies were from Promega (catalog nos. W4011 [rabbit], and W4021 [mouse]).

### Expression and purification of recombinant proteins

POLγA (wild-type and mutant variants), POLγB, and the TWINKLE DNA helicase were cloned and expressed as 6×His-tagged fusion proteins in Spodoptera frugiperda (Sf9) cells, using established protocols[38]. Due to its observed instability over time, the W748S variant was co-expressed with POLγB to maintain activity but was otherwise purified following the same protocol as the other variants. The mouse W726S variant also contained the E1121G polymorphism, and the human W748S variant the E1143G polymorphism. The gene encoding human or mouse mitochondrial single-stranded DNA binding protein (mtSSB1) were cloned into the pET-17b vector with a C-terminal 6×His-tag and purified following previously published methods[39]. Protein concentrations were determined by SDS-PAGE electrophoresis, using wild-type POLγA as a reference standard.

### DNA synthesis on ssDNA template

A $^{32}$P 5'-labeled 20-mer oligonucleotide (5'-GTA AAA CGA CGG CCA GTG CC-3') was hybridized to M13mp18 single-stranded DNA (New England Biolabs). Reactions were carried out in 20 μl volumes containing 0.5 nM template DNA, 25 mM Tris-HCl (pH 8.0), 1 mM TCEP, 10 mM $MgCl_2$, 35 mM NaCl, 0.1 mg/ml BSA, 200 nM human mtSSB (as a tetramer), 2.5 nM of the specified POLγA variants and 5 nM of the specified POLγB variants (calculated as a dimer) with indicating concentrations of all four dNTPs. Reactions were incubated at 37 °C for 20 min (45 min for Y933C) and stopped by the addition of 4 μl of stop buffer (90 mM EDTA, 6% SDS, 30% glycerol, 0.25% bromophenol blue and 0.25% xylene cyanol) and separated on a 0.8% agarose gel with 0.5 μg/mL EtBr at 40 V in 1 × TBE for 18 h. Products were visualized by

autoradiography. We conducted the experiments with human mtSSB and verified the results by repeating them with the mouse version of mtSSB (Supplementary Fig. 1b, c). For the mouse mtSSB experiments an additional Proteinase K (0.8 mg/ml) treatment at 42 °C for 30 min was conducted after stopping the DNA synthesis reaction, to prevent gel shift caused by the higher binding affinity of mouse mtSSB to the synthesis product. The time-course experiments were conducted under identical conditions, using a [32]P 5′-labeled 28-mer oligonucleotide (5′-ATC TCA GCG ATC TGT CTA TTT CGT TCA T-3′) hybridized to a single-stranded pBluescript SK(+) OriL template[38]. Reactions contained 5 nM POLγA and 7.5 nM POLγB (as a dimer) of the indicated variants and were terminated at the specified times. The 20-mer oligonucleotide hybridized to M13mp18 single-stranded DNA template described above was also used for SYBR Green assays but without 5′-radiolabeling of the primer. Reactions were performed at a final volume of 10 μL. Each reaction contained 0.5 nM of template DNA, 25 mM Tris–HCl (pH 8.0), 1 mM TCEP, 10 mM $MgCl_2$, 0.1 mg/mL BSA, 100 μM of all four dNTPs, 0.02% Triton X-100, 200 nM mtSSB (calculated as a tetramer), 0.5 nM of the specified POLγA variants, and 1 nM POLγB (calculated as a dimer). Reaction mixtures were distributed into microplates (384-well). The reactions were incubated at 37 °C for 1 h and terminated by the addition of 10 μl of 50 mM EDTA, 0.02% Triton X-100, and SYBR Green I (1: 5000) followed by incubation for 20 min at room temperature. The fluorescence signal was analyzed using a BMG PHERAstar microtiter plate reader (between 485–520 nm). The mean and standard deviations were calculated from 16 independent measurements.

## Processivity measurements

A [32]P 5′-labeled 20-mer oligonucleotide (5′- TGC GGG AGA AGC CTT TAT TT -3′) was hybridized to M13mp18 single-stranded DNA. The reaction consists of mixture A (10 μl), containing 50 mM Tris (pH 8.0), 0.2 mg/mL BSA, 2 mM TCEP, 50 mM NaCl, 5 nM template, 2.5 nM of the specified POLγA variants, 5 nM of the specified POLγB variants (calculated as a dimer) when indicated. The A mixtures were incubated for 10 min on ice prior to adding mixture B (10 μl) containing 20 mM $MgCl_2$, 20 μM of all four dNTPs and 600 ng/ml Heparin, when indicated. Immediately after mixing A and B (1:1) samples were put at 37 °C for 5 min and stopped by adding 20 μl stop buffer [95% Formamide, 20 mM EDTA (pH 8.0) and 0.1% bromophenol blue]. Samples were heated at 95 °C for 3 min before loaded on a pre-run (1 h, 1500 V) 8% denaturing PAGE-UREA [6 M Urea, 1 × TBE] sequencing gel and run for 75 min at 1500 V in 1 × TBE-buffer. The result was visualized in a Typhoon Imager (Cytiva).

Please note that the processivity assay was performed without mtSSB in the reaction, which differs from our DNA synthesis assay.

## Rolling circle in vitro replication assay

A 70-mer oligonucleotide (5′−42(T)ATC TCA GCG ATC TGT CTA TTT CGT TCA T −3) was hybridized to a single-stranded pBluescript SK(+) OriL followed by one cycle of polymerization using KOD polymerase (Novagen), as previously described[38], to produce a ~4-kb double-stranded template with a preformed replication fork. Reactions of 25 μl were carried out containing 0.4 nM template DNA, 25 mM Hepes (pH 7.6), 10 mM DTT, 10 mM $MgCl_2$, 35 mM NaCl, 0.1 mg/ml BSA, 1 mM ATP, 10 μM of all four dNTPs, 2 μCi [α-[32]P] dCTP, 160 nM human mtSSB, 8 nM human TWINKLE, and 4 nM POLγA with 8 nM POLγB for all variants except for the mixing experiments. In these experiments, the total amount of each individual POLγA variant was reduced by half, maintaining the same overall concentration by combining equal amounts of the indicated POLγA variants. For the experiments with Y955C/Y933C, 3 nM POLγA (either wild-type or mutant variant) and 6 nM POLγB (human or mouse, as indicated) were used instead. In these experiments, the ratio between wild-type and mutant variants was adjusted by keeping POLγA[WT] constant at 3 nM while increasing the concentration of the mutant variant as indicated. POLγB was increased proportionally along with the mutant POLγA variant. The 5-fold excess experiment was repeated using the same concentrations as in the Y955C/Y933C experiment, as indicated in Supplementary Fig. 2c, d. This experiment was performed with both the human and mouse versions of the recessive mutations. Reactions were incubated at 37 °C for the indicated times and stopped with 8 μl alkaline stop buffer (18% Ficoll, 300 mM NaOH, 60 mM EDTA (pH8), 0.25% bromophenol blue and 0.25% xylene cyanol). Products were run in 0.8% alkaline agarose gels at 40 V for 20 h and visualized by autoradiography.

## Exonuclease activity assay

To measure the 3′ to 5′ exonuclease activity of POLγA, a 20-mer oligonucleotide (5′-GCG GTC GAG TCC GGC GGC GC-3′) was [[32]P]-labeled at the 5′-end and annealed to a 36-mer oligonucleotide (5′-GAC TAC GTC TAT CCG GAC GCC GCC GGA CTC GAC CGC-3′), resulting in a 19-bp dsDNA region with a one-nucleotide mismatch at the 3′-end. Reaction mixtures (10 μl) consisted of 25 mM Tris-HCl (pH 8), 1 mM DTT, 10 mM $MgCl_2$, 0.1 mg/ml BSA, 8.7% glycerol, 1 nM template, 2 nM POLγA, and 4 nM POLγB of indicated variant, as specified in the figure. The reactions were incubated at 32 °C for the specified time (0–180 s), as indicated in Supplementary Fig. 13b. To stop the reaction, 2x stop buffer (98% formamide, 10 mM EDTA, 0.025% bromophenol blue, and 0.025% xylene cyanol) was added. The resulting products were analyzed by electrophoresis in 10% PAGE-UREA [7 M Urea, 1 × TBE] gels for ~2 h at 1500 V and visualized using autoradiography. Band intensities representing the 20-mer was quantified using Multi Gauge V3.1 software (Fujifilm Life Sciences). The fraction of non-degraded 20-mer was calculated by normalizing the background-subtracted intensities for each timepoint to that at $t = 0$. Standard deviations were calculated from three independent experiments to represent variability. The data were fitted to an exponential decay equation ("One phase decay") in Prism 10 (GraphPad Software) to determine the rate constants ($k_{exo}$) for human and mouse. Values are given with errors in s.e.m.

## The DNA binding activity

DNA binding affinity of POLγ to a primer-template was assayed using a 36-nucleotide (nt) oligonucleotide (5′-TTT TTT TTT TAT CCG GGC TCC TCT AGA CTC GAC CGC-3′) annealed to a [32]P 5′-labeled 21-nt complementary oligonucleotide (5′-GCG GTC GAG TCT AGA GGA GCC-3′). This produces a primed-template with a 15 bases single-stranded 5′-tail. Reactions were carried out in 15 μl volumes containing 0.5 nM DNA template, 25 mM Tris–HCl (pH 8.0), ~30 mM NaCl, 1 mM TCEP, 0.1 mg/ml BSA, 10% glycerol, 10 μM ddCTP. POLγB (6.6 nM, calculated as a dimer) was included in the mixture, and POLγA (0, 0.01, 0.02, 0.04, 0.08, 0.17, 0.33, 0.67, 1.33 nM) was added as indicated in the figures. Reactions were first incubated on ice for 10 min, followed by an additional 10 min at room temperature. Samples were then separated on a 6% native PAGE gel in 0.5 × TBE buffer and electrophoresed for 40 min at 180 V. Bands were visualized using autoradiography.

For $K_d$ analysis, band intensities corresponding to unbound and bound DNA were quantified using Multi Gauge V3.0 software (Fujifilm Life Sciences). The fraction of bound DNA was determined from background-subtracted signal intensities using the formula: bound/(bound + unbound). The fraction of DNA bound in each reaction was plotted against the concentration of POLγ. Data were fitted using the quadratic equation: (Fraction DNA bound = $(([POL\gamma]_{tot} + [DNA]_{tot} + K_d) - sqrt(([POL\gamma]_{tot} + [DNA]_{tot} + K_d)^2 - 4[POL\gamma]_{tot}[DNA]_{tot}))/2[DNA]_{tot}$) in Prism 10 (GraphPad Software) to obtain $K_d$ values. Each $K_d$ value represents the average of three independent reactions.

## Steady state kinetics

The template used for the steady state kinetics is the M13 ssDNA template, the same as that used in the DNA synthesis assay on ssDNA but without 5′-radiolabeling. For $K_{m\_app, dNTP}$ determination, reactions were performed in 160 µl volumes containing 1 nM of the DNA template, 25 mM Tris−HCl (pH 8), 1 mM TCEP, 25 mM NaCl, 10 mM MgCl$_2$, 0.1 mg/ml BSA, 1.3 nM POLγA, 3.75 nM of POLγB (as a dimer), 200 nM of tetramer mtSSB, and the indicated concentrations of the four dNTPs. For the determination of steady state parameters for the reactions of POLγA alone, the same conditions were used but at 2.6 nM POLγA without presence of neither POLγB nor mtSSB.

The dNTPs were spiked with a small amount of $\alpha^{32}$P-dCTP in order to follow nucleotide incorporation. Typically, 40 µCi ≈ 13 pmol was added to 100 µl 200 µM dNTP solution. This dNTP solution was then diluted accordingly, both for the reactions and the calibration curve. The reactions were incubated at 37 °C and 20 µl samples were taken at 0, 2.5, 5, 7.5, 10 and 12.5 min and stopped by adding 5 µl of 0.5 M EDTA. The products were analyzed by placing a 5 µl droplet onto a Hybond N$^+$ positively charged membrane (# RPN303B, Amersham Cytiva). The membrane was air-dried 15 min and then washed to remove non-incorporated nucleotides 3 × 5 min in 2 × SSC followed by a 2 min wash in 99.5% Ethanol, air-dried again and visualized by autoradiography. The standard curve was generated by adding 5 µl of each indicated amount of dNTP on a separate membrane. This membrane was dried without washing representing total amount of dNTP. The dots were quantified using Fujifilm Multi Gauge V3.1 software and initial rates were determined for each [dNTP]. Velocity was plotted (using Prism 10) against [dNTP] and non-linear regressions using "Michaelis-Menten" algorithm were performed to get values for $K_{m\_app, dNTP}$ and $k_{cat}$. Reactions were performed at least three times.

## Off-rate measurements

For off-rate measurements, POLγ binding to a primer-template was assayed using a 40-mer oligonucleotide (5′-TTT TTT TTT TAT CCG GGC TTC TCT AGA CTC GAC CGC ATG C-3′) annealed to a complementary 25-mer oligonucleotide (5′-GCA TGC GGT CGA GTC TAG AGG AGC C-3′) at a 1:1 ratio, with 10% of the 25-mer oligonucleotide 5′-$^{32}$P-labeled.

Human or mouse POLγA$^{WT}$ and POLγB$^{WT}$ (calculated as a dimer) were mixed at a 1:1.5 ratio and pre-incubated on ice for 5 min. The reaction mixture was then prepared by adding the pre-incubated protein mix to mixture A (50 mM Tris (pH 8.0), 2 mM TCEP (pH 8.0), 0.2 mg/mL BSA, 40 mM NaCl, 2% DMSO). The DNA template was added to mixture A before an equal volume of mixture B (20 mM MgCl$_2$, 40 µM dCTP) was added. The final reaction conditions included 50 nM DNA template, ~30 mM NaCl, 1 nM POLγA, and 1.5 nM POLγB. The reactions were immediately put at 37 °C and every 15 s 20 µl of the reaction was withdrawn and immediately quenched by mixing with 20 µl of stop buffer (95% Formamide, 100 mM EDTA (pH 8.0) and 0.1% Bromophenol blue). Samples were loaded onto a 10% denaturing PAGE-UREA [7 M Urea, 1 × TBE] sequencing gel and run for 1 h and 45 min at 1500 V in 1 × TBE-buffer. The result was visualized by Phosphor Imaging. Band intensities representing 25- and 26-mer were quantified using Multi Gauge V3.0 software (Fujifilm Life Sciences). The fraction of synthesized [γ-$^{32}$P] ATP-labeled 26-mer was determined using background-subtracted signal intensities and the expression: 26-mer/(25-mer + 26-mer) for each sample. The fraction [γ-$^{32}$P] ATP-labeled 26-mer synthesized was converted to nM by multiplying with the concentration of labeled template and then multiplied by 10 to get the total concentration of 26-mer synthesized in the reaction. 26-mer synthesized (nM) was plotted against time (s) and the data was fit using "simple linear regression" in Prism 10 (Graphpad Software). To obtain the dissociation rate constant $k_{off}$, the slope of the linear regression was divided by the concentration of POLγA in the reaction. The obtained $k_{off}$ (s$^{-1}$) for each of the replicates were plotted as individual experiments in a scatterplot with indicated mean and errors shown as s.d.

## Cryo-electron microscopy sample preparation and data acquisition

Primer-template DNA with a single 3′ mismatch was formed by annealing a 25-nt primer (5′-GCA TGC GGT CGA GTC TAG AGG AGC T-3′) to a 40-nt template strand (5′-TTT TTT TTT TAT CCG GGC TCC TCT AGA CTC GAC CGC ATG C-3′). To prepare cryo-EM samples, mPolyA or hPOLγA was mixed at a 1:2 molar ratio with mPolyB or hPOLγB and dialyzed into a buffer containing 20 mM HEPES−NaOH (pH 7.5), 140 mM KCl, 10 mM CaCl$_2$, and 1 mM TCEP. The DNA substrate was added at a POLγ:DNA molar ratio of 1:1.2, and dCTP was added to a final concentration of 0.2 mM. The samples were incubated on ice for 10 min before being applied to the grids. Grids were prepared using a Vitrobot Mark IV (Thermo Fisher Scientific) by adding 3.5 µl of approximately 2 µM protein sample to glow-discharged QuantiFoil 2/1 grids (Quantifoil Micro Tools GmbH) at 4 °C and 100% humidity. Grids were blotted for 2 s and plunge-frozen in liquid ethane.

Cryo-EM data were collected on a Titan Krios G2 microscope (Thermo Fisher Scientific) at the SciLifeLab cryo-EM facility in Stockholm, operated at 300 kV and equipped with a K3 direct electron detector (Gatan) and a BioQuantum energy filter (Gatan) set to a slit width of 20 eV. Cryo-EM data was collected in super-resolution mode at a nominal magnification of ×105,000 and then 2× binned, corresponding to a calibrated pixel size of 0.825 Å (0.828 Å for the collection of chimeric complexes). The total dose was 40 $e^-$ per Å$^2$ per 40 frames, and the movies were acquired at a defocus range of −0.8 µm to −2.2 µm.

## Cryo-electron microscopy data processing

The processing workflows are shown in Supplementary Figs. 3, 7, 9. The acquired movie stacks were imported into cryoSPARC (v4.3.1)[40] for image processing. Motion correction of the data was performed with Patch Motion Correction, and the contrast transfer function (CTF) was estimated with Patch CTF Estimation. Micrographs with poor CTF fit (worse than 4 Å) were removed, and the Automatic Blob Picker was used to pick particles, which were extracted with 4× binning (3.3 Å per pixel). Multiple rounds of two-dimensional (2D) classifications were performed to filter out junk particles. The remaining particles were used to perform ab initio reconstructions followed by heterogeneous refinement.

For mPoly, three classes displayed clear POLγ features in different conformations. All three classes were kept and filtered further with several rounds of 2D classifications and heterogeneous refinements with one decoy volume to remove junk particles. The particles were re-extracted without binning (0.825 Å per pixel) and each of the three volume classes were refined with non-uniform refinement. At this stage, the R, I and E conformers were all discernable. The particles in the R conformer processed further with CTF refinements and a final round of non-uniform refinement, resulting in a map gold-standard FSC (GSFSC) resolution of 2.75 Å. The I conformer class was divided into two subclasses after a round of heterogenous refinement. The first class had clear features of the I conformer and after CTF refinements and non-uniform refinement a 2.98 Å map was obtained. The E conformer class was filtered further with a three-class heterogenous refinement. The second class was merged with the second subclass from the I conformer heterogenous refinement, as this class also showed clear E conformer features. CTF refinements and non-uniform refinement yielded a 2.87 Å map. To improve the details of the E conformer, two local refinement jobs were run with masks created to cover either the POLγB subunits or the POLγA and the DNA. Local maps were obtained at 2.77 Å and 2.70 Å resolution for POLγA + DNA and the POLγB subunits, respectively.

For the mPolγA/hPOLγB chimera, four classes displayed clear POLγ features in different conformations. These classes were filtered further with several rounds of 2D classifications and heterogeneous refinements with one decoy volume to remove junk particles. The particles were re-extracted without binning (0.828 Å per pixel) and refined with non-uniform refinement. At this stage, the R, I and two E conformers were observed. The particles in the R and I conformers were processed further with CTF refinements and a final round of non-uniform refinement, resulting in map gold-standard FSC (GSFSC) resolutions of 2.54 and 2.80 Å, respectively. The E conformer classes were filtered further with a three-class heterogenous refinement. One class lacked DNA in the exonuclease site and was discarded. The remaining two classes were mouse-like or human-like E conformers. The mouse-like E conformer underwent another round of a 2-class heterogenous refinement. One of these classes, along with the human-like E conformer, were processed with CTF refinements and non-uniform refinement to yield 3.24 Å and 3.08 Å maps, respectively. To improve the details of the E conformers, local refinement jobs were run with masks created to cover either the POLγB subunits or the POLγA and the DNA. For the mouse-like E conformer, local maps were obtained at 3.09 Å and 3.19 Å resolution for POLγA + DNA and the POLγB subunits, respectively. For the human-like E conformer, local maps were obtained at 3.02 Å and 2.94 Å resolution for POLγA + DNA and the POLγB subunits, respectively.

For the hPOLγA/mPOLγB chimera, three classes displayed clear POLγ features in different conformations. These classes were filtered further with several rounds of 2D classifications and heterogeneous refinements with one decoy volume to remove junk particles. The particles were re-extracted without binning (0.828 Å per pixel) and refined with non-uniform refinement. One class was used for heterogeneous refinement and followed by non-uniform refinements and CTF refinements, resulting in R and I conformer maps gold-standard FSC (GSFSC) resolutions of 2.73 and 2.91 Å, respectively. Maps with E conformer features underwent multiple rounds of heterogenous refinements to filter out classes without DNA in the exonuclease site. Two classes were combined and were processed with CTF refinements and non-uniform refinement to yield a 2.96 Å map of a mouse-like E conformer. To improve the map, local refinement jobs were run with masks created to cover either the POLγB subunits or the POLγA and the DNA. Local maps were obtained at 2.87 Å and 2.87 Å resolution for POLγA + DNA and the POLγB subunits, respectively.

The GSFSC = 0.143 criterion was used for determining the resolution of all reconstructions. All final maps were sharpened using DeepEMhancer or Phenix autosharpen[41,42]. For the E conformers, composite maps were generated by docking the local maps in the consensus map and then using the volume maximum command in ChimeraX[43].

### Model building, refinement, and analysis

To build the mPolγ structures, the hPOLγ ternary complex (PDB ID: 4ZTZ) was docked into the cryo-EM maps by rigid body fitting in UCSF ChimeraX (v.1.4), and then mutated to match the mPolγ^WT amino acid sequence and manually fitted in real space in Coot (v.0.9.8.1) and ISOLDE (v.1.4)[43–45]. The same strategy was used for the chimeric complexes, but the editing conformer of hPOLγ^WT (PDB ID: 8D42)[24] was used as a starting point for the hPOLγA^WT subunit in the hAmB complex. Model building was not performed for the chimeric I conformers. Refinement and validation statistics, as well as information about the cryo-EM data collection, are detailed in Supplementary Table 1. After initial fitting, the models were improved iteratively with real-space refinement in PHENIX and manual adjustments in Coot[42]. The maps sharpened in Phenix were used as primary maps, and the DeepEMhancer maps were used to facilitate interpretation and tracing. The refined models were validated with MolProbity against the maps sharpened in Phenix (Supplementary Table 1)[46]. Local resolution was

estimated for the consensus maps with cryoSPARC (FSC threshold = 0.5) (Supplementary Figs. 4, 8, 10). All figures were prepared using UCSF ChimeraX.

### Thermal shift assay

Thermal shift assays performed with SYPRO Orange dye (Invitrogen) were used to obtain normalized fluorescence melting curves to determine the apparent melting temperatures of POLγA, POLγB or the POLγ complex. Mouse or human proteins were mixed and incubated on ice for 5 min before adding the reaction mixture (20 mM Tris (pH 8), 75 mM NaCl, 1 mM DTT, 5x SYPRO Orange dye). Three replicates of 20 µl reactions, each containing 0.5 µM mPolγA with or without 0.75 µM mPolγB (1:1.5 ratio), were transferred to the 96-well plates (final NaCl concentration was 175 ± 50 mM). Temperature-induced unfolding was performed by a stepwise increase in temperature (0.5 °C every 5 s), from 4 °C to 95 °C. The fluorescence signal of the SYPRO Orange dye was detected using the HEX emission filter (560–580 nm) on a Bio-Rad CFX Opus 96 machine. The melting temperatures were determined from the first derivate of the plot of fluorescence intensity versus temperature. Standard deviation was calculated from three independent experiments ($n = 3$).

### Statistical analysis

All numerical data are expressed as mean ± s.e.m. unless otherwise stated. A two tailed Student's t-test was used to assess statistical significance (see figure legends for details) in two groups comparisons. For multiple comparisons in Fig. 1e, f, Welch and Brown-Forsythe one-way ANOVA was used. Two-way ANOVA test with Tukey's correction was used for multiple comparisons. Differences were considered statistically significant for $p < 0.05$. Prism Graphpad v9 and 10 have been used for statistical analysis. No statistical method was used to predetermine sample size, and group sizes were determined based on the results of preliminary experiments. No randomization was used, and animals were assigned to the experimental groups based on the genotype. No blinding to the operator was used.

### Reporting summary

Further information on research design is available in the Nature Portfolio Reporting Summary linked to this article.

## Data availability

The atomic models and cryo-EM density maps have been deposited in the Protein Data Bank and the Electron Microscopy Data Bank under the following accession codes: Mouse R conformer (9G74, EMD-51109), Mouse I conformer (9G75, EMD-51110) and the Mouse E conformer (9G77, EMD-51114). Accession codes for consensus and local refinement maps of the mouse E conformer are EMD-51111 (consensus), EMD-51112 (local refinement of mPolγA and DNA) and EMD-51113 (local refinement of mPolγB subunits). The chimeric complex (mAhB) was assigned the following accession codes: R conformer (9IBX, EMD-52815), human-like E conformer (9IBZ, EMD-52819) and mouse-like E conformer (9IC0, EMD-52823). Accession codes for consensus and local refinement maps of the mAhB human-like E conformer are EMD-52816 (consensus), EMD-52817 (local refinement of mPolγA and DNA) and EMD-52818 (local refinement of mPolγB subunits). Accession codes for consensus and local refinement maps of the mAhB mouse-like E conformer are EMD-52820 (consensus), EMD-52821 (local refinement of mPolγA and DNA) and EMD-52822 (local refinement of mPolγB subunits). The chimeric complex (hAmB) was given the following accession codes: R conformer (9IC1, EMD-52824) and mouse-like E conformer (9IC3, EMD-52828). Accession codes for consensus and local refinement maps of the hAmB mouse-like E conformer are EMD-52825 (consensus), EMD-52826 (local refinement of mPolγA and DNA) and EMD-52827 (local refinement of mPolγB subunits). The human R conformer (8D37, the human I conformer (8D3R, and the

Human E conformer (8D42 were used for comparison. Source data are provided with this paper.

## Materials availability
Plasmids and mouse models generated in this study are available from the authors upon request.

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

## Acknowledgements

Work described here was supported by the Swedish Research Council (2021-00932 to M.F., 2022-00976 to C.M.G.), the Swedish Cancer Foundation (to M.F. and C.M.G.), the Knut and Alice Wallenberg foundation (to M.F.), Telethon Foundation (GGP20013 and GSP24003 to C.V., and GMR23T1065 to M.Z.), AFM-Téléthon (23706 to C.V.), Associazione Luigi Comini Onlus (MitoFight2, to C.V. and M.M.), Department of Biomedical Sciences—UNIPD; The Lily and POLG Foundation Awards (to M.Z. and C.V.); The Champ Foundation grant (to P.S.P. and M.M.); PNRR Mission 4, Component 2, Investment 1.4-CN00000041 Spoke 1 funded by the European Union—NextGenerationEU, the Medical Research Council (UK) awards MC_UU_00015/4 and MC_UU_00028/3 (to M.M.) and MC_PC_21046 to establish a National Mouse Genetics Network Mitochondria Cluster (MitoCluster) (to C.V. and M.M.). We thank Dr Jay P. Uhler for help with preparing illustrations. We also thank Dr. Jens Berndtsson and the Centre for Cellular Imaging at the University of Gothenburg, the National Microscopy Infrastructure (VR-RFI 2019-00217), Dr. Marta Carroni, Dr. K. Walldén at SciLifeLab Stockholm for helping with cryo-EM microscopy. The data were collected at the Cryo-EM Swedish National Facility funded by the Knut and Alice Wallenberg, Family Erling Persson and Kempe Foundations, SciLifeLab, Stockholm University and Umeå University. We are grateful to: Simon Gillard, Michelle Sandell, Tilly Passmore, Clare Norris and the Harwell phenotyping and data teams, Eloisa Turco, Federico Caicci, Sara Schiavon and Francesco Boldrin of the Imaging Facility at DiBIO, University of Padova.

## Author contributions

C.V., M.Z., X.Z. and M.F. designed the study, S.C., A.Z., R.C., S.Vo. and H.D. performed the experiments on mouse models, P.S.P. designed the CRISPR/Cas9 strategy to create the mouse models, L.J., E.H., K.A.S.J., S.S., M.F., S.C., S.Va. and B.M. performed the in vitro experiments, S.Va. and U.R. performed the structural studies, A.T. and A.S. contributed critical reagents, M.M. and C.G. contributed to the interpretation of the data. All the authors contributed to the manuscript.

## Competing interests

M.F., C.M.G. and M.M. are co-founders. M.F., C.M.G., B.M., and X.Z. are shareholders of Pretzel Therapeutics and have received consulting fees. S.V. has received consulting fees from Pretzel therapeutics. The other authors do not have COI to disclose.
