## [Transparent Peer Review file · Nature Communications]

Modelling POLG mutations in mice unravels a critical role of POLyB in regulating phenotypic severity

Corresponding Author: Professor Carlo Viscomi

Version 0:

Reviewer comments:

Reviewer #1

(Remarks to the Author)

The manuscript by Corra et al investigated the feasibility of using mouse model to study human mitochondrial DNA polymerase mutations implication in diseases. The authors performed a commendable series of experiments of molecular biology, structural biology, enzyme kinetics, and animal studies. As many as 300 POLG mutations have been implicated in human diseases, yet their causality of the diseases has not been evaluated independently. The work investigated four most common human POLG mutations and concluded that mouse can be a valid model for validation of mutations in the human counterpart. The work has significant impact on development of therapeutic intervention to human mitochondrial diseases. The paper is clearly written, and data are of high quality. However, the data analyses and implication could be improved in Results and Discussion and some errors need to be corrected prior to publication.

1. Mouse Polg mutants activates shown in Fig. 1. mW726S (Fig. 1D) is more defective than mG826S (1E), yet in the summary (1H) their activities are opposite. Explain.

2. Structural analyses: it is wonderful the authors successfully solved cryo-EM structures of mouse Pol g holoenzyme. As mouse mutation correspond to the human A457T did not cause dysfunction as the mutants, and the authors have mapped out the sequence difference between human and mouse protein, analyses based on structural difference will be informative in understanding what types of mutants of human Pol g mutations are suitable for mouse studies.

3. Related to the above point, what amino acid(s) might contribute to the difference in E and R conformers transition in human and mouse enzyme?

4. Enzyme kinetics (Fig. 4H): due to the processivity of the polymerase, k_{cat} (V_{max}) for the holoenzymes is dominated by k_{off} of enzyme from DNA, not the catalytic rate. 17-33/min are much slower than the polymerization rate measured by the presteady state rate. Thus, the faster k_{cat} for the mouse Polg may not reflect superior catalytic activity but faster off-rate. It is interesting though the $KM(dNTP)$ for mPol g is lower than hPolg, but hPolg has tighter affinity to DN A(Fig.S6D), can these be rationalized from the structures?

5. Authors should speculate the structural basis for chimera holoenzymes, hAmB and mAhB activities. What would the subunit interface for these chimera enzymes look like, more human- or more mouse-like?

6. Molecular weight standard should be included in gels in Figure 2, as well as the lengths of 'T' and 'P' in panel A. Also, information on mtSBB and mTwinkle protein cloning and purification. was missing.

7. Fig S6D: hPOLg DNA binding curve does not fit the actual data, data fitting should be re-performed, the K_d for human should be tighter than reported.

8. Data in Fig S6G and S7A appear to be inconsistent: in Fig S6G, the hAhB did not produce long product up to 10 min whereas mAmB produced longer product in 5min and nearly extended all primers to FL in 10 min. However, in Fig. S7A, hPolg performed similarly if not better than mPolg in 5min. What could be the reason for the discrepancy?

9. Animal studies more description on animal behaviors.

a. Treadmill was listed in the method, but no result of the test is presented.

-Fig. 5 & S11

a. Rationalize why PolgA449T/KO is more defective than Polg A449T/A449T

b. What would be the basis for the tissue specificity for OXPHOS activity and the complex specificity, i.e., Fig. S11A, Polg A449T/A449T appears cause more damage to CI than CIV in SKM, but equally defective in Liver and have no effect on either in Brain.

-Fig. S12

How do these data compare to the same animals at 24th month?

Minor points

10. Line 80: reference does not correspond to the statement in that paragraph.

11. POLg, Polg and Polg were used interchangeably, please unify the terminology.

12. Figure 4: panel A is not in the figure legend.

13. line 338: suggest change "Based on the mtDNA data" to "Based on the mtDNA copy number"

14. unify 'in vitro' and 'in vitro'.

15. line 616: add the full stop at the end of "...recombination as described in¹⁰".

16. line 630, delete '10.'

17. line 632: missing a punctuation at the end of "...Figure S8B"

18. superscript 32 in '32P' through the text

19. use prime for 5 and 3 ends through text (example 5' to 3')

20. line 738: use the correct format for temperature, for example: 37°C.

21. line 745: delete the – from -3' - in oligo sequence.

22. line 751: change 0,1% to 0.1%

23. line 757: "A 70-mer oligonucleotide (5'-42(T)-ATC TCA GCG ATC TGT CTA TTT CGT TCA T-3') ..." is 42(T) a typo?

24. line 776: please use the subscript for MgCl₂.

25. line 798-799: Change KD to KD

Reviewer #2

(Remarks to the Author)

Reviewer #3

(Remarks to the Author)

Corrà et al. report generation of mutant mice to model the most commonly found disease-linked mutations in the catalytic subunit of human mitochondrial DNA polymerase, PolγA. Both homozygous and compound heterozygous mutants were made. The recombinant forms of these mutants were purified and were tested for their catalysis of DNA synthesis on single-stranded and double-stranded DNA templates. In general, the wild-type and mutant mouse Polγ performed better at DNA synthesis than the corresponding human Polγ variants. Authors claim that these differences arise due to differences in mouse and human PolγB and their interactions with their corresponding PolγA subunits. The mutant mice were tested for their neurobehavior, tissue composition, mtDNA copy number etc., with more in-depth analysis of autosomal dominant Polγ+/Y933C mice. The strength of the study lies in establishing mice models. These diseases have no cure, and the animal models could be of great help in the future to understand the mechanisms leading to the diseases as well as to screen therapeutic interventions against them.

Overall, this study makes an important contribution to the field as PolgAB mutants have been known for a long time, but the disease manifestations have not been studied well. However, the manuscript in its current form substantially lacks at dissecting the mechanisms leading to the enhanced activities of the mouse PolgAB and mutants. The structures of mouse and human PolgAB are very similar. The basis for the higher activity of mouse PolgAB is not entirely clear. There are other explanations as indicated in the comments below. This restricts the claimed 'genotype-activity-phenotype' correlation (page 6) to reduced DNA synthesis by the mutants and does not provide mechanistic basis for the highly variable phenotypes observed for the different mutants. The discussion should include a model that explains how the authors think the mouse PolgB has an advantage over the human PolgB and how it could alleviate the defects from the specific PolgA mutations.

Specific comments:

1. The in vitro primer extension reactions with mouse DNA Poly have been performed with human mtSSB, and not with the mouse mtSSB. The higher activity of mouse PolgAB on the DNA substrates could be due to the human mtSSB or TWINKLE used in the assays. This needs to be experimentally addressed.

2. In general the mouse PolgAB mutants have slightly higher activities than human PolgAB. The study does not nail down why this is the case. PolgB is a processivity factor. Does mouse PolgAB have higher processivity or rate of DNA synthesis or lower exonuclease activity? Such mechanistic analyses or discussion is missing here.

3. The velocity in figure 4F reported for the mouse PolgAB is higher. This could be due to better assembly of the mouse complex on the DNA substrate rather than higher processivity or rate of DNA synthesis.
4. The differences in the exonuclease activity of mutant mouse Poly and human Poly could be responsible for the different activity which is not compared here.
5. From the gels in Figures 1 and 2, it appears that using 1:1 ratio of wild-type Poly and mutants recovers the defects produced by the mutants alone. Interestingly, this appears to be the case for both recessive and dominant mutants. What are the mechanistic implications of this observation? Increasing the proportion of the Y955C (or mouse Y933C) mutant from 1:1 to 5:1 reduces the DNA synthesis. How would the recessive mutants fair under these conditions?
6. Line 118: Related to the claim "Using cryoelectron microscopy (cryo-EM), we establish the structure basis of modulating Poly activity by disease causing mutations in mouse".
Cryo-EM structures are used to understand the interactions between PolyA and PolyB in mouse and human proteins to explain the differences observed between the polymerase activities in the two models. Structures of human (or mouse) PolyA mutants under focus have not been presented. Moreover, except for the mutant A449T (A467T in human), none of the other mutants have been discussed in relation to the cryo-EM structures presented in the manuscript. Thus, the claim to the structural basis of the activities of the disease-causing mutations is not well substantiated.
7. Supplementary Fig. S6: There are two bands representing Poly Bound to Primer-template, specially at the higher protein concentrations. Could the authors speculate the origin of the two bands?
8. Supplementary Fig. S1. "Mutations impair polymerase activity of the human POLy across all dNTP concentrations tested". The title is incorrect as different lanes show a time-course at a constant dNTP concentration.

Reviewer #4

(Remarks to the Author)

Reviewer #5

(Remarks to the Author)

Review of the manuscript "Modelling human disease-causing POLG mutations in mice unravels a critical role of POLyB in regulating the phenotype severity" by Corrà et al.

The manuscript offers insights into how POLG mutations affect mitochondrial DNA replication in humans and mice, with implications for understanding disease mechanisms and developing therapies. The most common POLG mutations (A467T, W748S, G848S, Y955C) were analysed with in vitro assays, and mouse models. DNA synthesis assays show how the mutations impair polymerase activity in both human and mice proteins, with mouse proteins generally less affected, but still respecting the severity observed for the human counterparts. The cryo-EM structure of the mouse POLy shows differences with the human POLy, particularly within the POLyB subunit and the AID domain, that show less flexibility in the transition from the polymerase mode to the editing mode, potentially explaining the milder mutation effects in mice. Despite the absence of mtDNA deletions, the mouse models demonstrates that the pathogenetic mechanisms are largely conserved, highlighting the value of these models for understanding POLy-related diseases and developing new therapies.

This is an elegant manuscript, where by application of complementary approaches the authors bring a new tool to investigate the effect of common POLG mutations. The experiments are well done, the text is very compact, understandable and logically arranged.

Minor comments:

1- "These mutations are located in different regions of the POLyA protein. The A467T mutation is situated in the thumb region of the polymerase domain, while the W748S mutation is in the linker region that connects the proofreading and polymerase domains, and is important for interactions between POLyA and POLyB."

Would be nice to have a schematic showing the position of these mutations on the protein structure

2- Figure 1 and Figure 2

It is true there is a milder effect of some mutations over others. But given the small differences, to make a proper comparison among mutants, an SDS page before dilution to final concentration should be provided showing that comparable amount of protein was used in the assay. It would also be good to have an idea of how these mutations affect the overall stability of the protein, maybe by looking at the thermal stability.

4- "Clearly, mPolyB contributes to the differences in activity observed between the human and mouse POLy holoenzymes.

The effects of mutations are similar for hPOLyA and mPolyA in isolation, but when combined with their accessory factors, the stronger stimulatory effect of mPolyB partially conceals the negative effects of disease-causing mutations in the mouse compared to the human system”

It is clear how POLyB stimulates DNA synthesis, and mPolyB has a stronger effect than the human counterpart. The structural observations suggest that the more flexible human proteins (within AID and POLyB domains) are the reason for this difference. To strengthen this hypothesis a set of complementary DNA binding assays would be useful. How does the affinity for the DNA changes in presence and absence of POLyB? Does mPolyB increase the affinity of hPOLyA for DNA?

5- In the EMSA shown in figure S6.C there is a double shift at high concentration. The reason for this was not explained in the text. A different assay like fluorescence anisotropy with a labelled DNA template could be more suitable in picking up small differences in affinity to the DNA.

3- Figure S7B:

Although the mutations have different numbers between human and mouse, it would still be visually helpful to have the “h” or “m” in Figure S7B in front of the mutated residues

6- W748S>G848S>A467T>Y933C

As the mutations with higher severity are on the right, the symbol should be inverted: < instead of >

7- “Taken together, our findings point to POLyB as a potential therapeutic target, which could be leveraged to mitigate the effects of various POLy mutations.”

While it is clear how the POLyB subunit enhances the polymerase activity, remains undiscussed how this subunit could be used as a potential therapeutic target

Version 1:

Reviewer comments:

Reviewer #1

(Remarks to the Author)

The authors have thoroughly addressed our critiques. The revised manuscript include new data on thermostability measurements of Pol gamma variants, two new cryo-EM structures and repeated enzymatic analyses. We are satisfied with the revision and believed its has met the requirement for publication.

Minor comments:

1. Reference #40 appears to be incorrect.
2. Double check the Kd value presented in Sup Fig.13f, which is three-orders of magnitudes higher than the published value.

Reviewer #2

(Remarks to the Author)

Reviewer #3

(Remarks to the Author)

Authors have satisfactorily addressed all the comments and concerns raised, improving the manuscript considerably. I have some minor comments:

1. Line 198: “This observation aligns with the fact that dominant mutations are relatively mild, leading to mtDNA deletions rather than depletion in affected patients”. Please cite relevant references supporting the statement.

2. Line 113: “Both mouse PolyA and PolyB are conserved in most mammals, with mPolyA sharing 85% sequence identity with its human counterpart”. I think the authors want to convey that both PolyA and PolyB are conserved among most mammals, rather than ‘mouse’ PolyA and PolyB are conserved.

3. Authors should note in the methods that the processivity experiments were performed without mtSSB to dissuade the readers from comparing DNA syntheses in Figure 1b and Supplementary Figure 14a.

4. Line 210: “To better understand why mutations generally cause a less severe molecular phenotype in mouse vs human Poly, we employed single-particle cryogenic electron microscopy to determine the structure of the mPolyWT-DNA complex”. As no cryoEM structures of the mutants are reported, the structures presented inform more on the differences between mPolyWT and hPOLyWT rather than explaining less severe phenotypes observed in mouse Poly mutants.

5. Figure 6. It will be helpful to add H & E, COX/SDH etc. on the figure panels.

6. Error bars in Figures 1e, 1f, 4h, 4i and 4j should be defined in the respective figure legends.

7. Line 779: "The template used for the steady state kinetics is the same that was used in the DNA synthesis assay on ssDNA but without 5'-radiolabeling". There are two templates that were used for DNA synthesis on ssDNA (M13 ssDNA and single-stranded pBluescript SK(+) OriL). Specify which one was used.

Reviewer #4

(Remarks to the Author)

Reviewer #5

(Remarks to the Author)

The authors have comprehensively addressed all comments with clarity and rigor. The revised manuscript shows significant improvements and meets the standards expected for publication.

**Responses to referees' comments – Nature commun manuscript 24-53797:
"Modelling POLG mutations in mice unravels a critical role of POL γ B in regulating phenotypic severity"**

Response to Reviewer #1

We would like to thank the reviewer for their detailed and thoughtful feedback. We have addressed each of the points below and have made revisions to the manuscript to incorporate the suggestions and clarify any points of confusion.

Referee #1, comment 1. Mouse Polg mutants' activities shown in Fig. 1. mW726S (Fig. 1D) is more defective than mG826S (Fig. 1E), yet in the summary (Fig. 1H) their activities are opposite. Explain.

Our Response:

We thank the reviewer for pointing out this inconsistency in our data. As noted, in the earlier version of the manuscript, the W726S mutant appeared more affected than expected in Figure 1D compared to Figure 1H. The reviewer's comment prompted us to re-quantify all POLG variants used in our study.

Interestingly, the concentrations of all POLG variants, except for W726S, were very similar to those measured a year ago when the experiments were performed. However, the W726S variant was only about 50% of the expected concentration, suggesting that its levels were decreasing over time, potentially due to precipitation during storage.

Although we always aliquot the proteins and never reuse aliquots, a protein prone to precipitation may still exhibit variation between tubes, which could explain the observed discrepancy in the figures. To address this, we repeated all relevant experiments using newly determined protein concentrations and updated the manuscript accordingly. The revised figures (Figure 1b, c, e and f and 2b-c) and now include these new experiments, and the inconsistency noted by the reviewer is no longer apparent.

*To ensure that comparable protein levels were used across all experiments, we also confirmed the measured protein concentrations by separating and visualizing the POLG variants on an SDS-PAGE gel (see **Supplementary Fig. 1a**).*

*Additionally, to investigate potential stability issues, we performed thermal shift assays on all mouse variants in the presence and absence of POLGB (see **Supplementary Fig. 12**). Interestingly, W726S did not exhibit lower thermostability than the other mutants in this assay, indicating that the observed decrease in protein concentration over time is not due to inherent thermostability issues, but may be influenced by other physicochemical properties of the protein.*

Referee #1, comment 2. Structural analyses: it is wonderful the authors successfully solved cryo-EM structures of mouse Pol g holoenzyme. As mouse mutation correspond to the human A457T did not cause dysfunction as the mutants, and the authors have mapped out the sequence difference between human and mouse protein, analyses based on structural difference will be informative in understanding what types of mutants of human Pol g mutations are suitable for mouse studies.

Our Response:

We appreciate reviewer's valuable comments. We have expanded our discussion on the structural differences, or rather similarities, between human and mouse Pol γ and their implications for modeling human mutations in mice. Specifically, we highlight that the four mutation sites (A449T, W726S, G826S and Y933C) are close to identical between mouse and human (in all three conformers). In fact, most residues in the catalytic subunit are conserved, both in sequence and structurally, meaning that most POLG mutants are likely comparable between mouse and human (although the effect always seems to be milder in mouse).

Interestingly, our structural analyses revealed a distinct positioning of the POL γ B subunit in the error-editing conformer of the mouse enzyme compared to the human enzyme. This difference may contribute to the higher activity of mouse POL γ and the generally milder effects of mutations in mice, suggesting that POL γ B plays a key role in modulating the functional impact of POLG mutations between species. Given this, it may be that exchanging POL γ B, rather than POL γ A, could recreate the more severe human phenotypes in mouse models.

Referee #1, comment 3. Related to the above point, what amino acid(s) might contribute to the difference in E and R conformers transition in human and mouse enzyme?

Our Response:

*Our data suggests that POL γ B plays a major role in the different behavior observed between mouse and human, and it is therefore likely that amino acids involved in the interaction between catalytic and accessory subunits are responsible. However, most substitutions are also in these regions, making it difficult to pinpoint the exact residues involved. Nevertheless, we believe that the residues in the L-helix are good candidates as this helix harbors many substitutions and are directly involved in the interaction with POL γ B. Related to this, we have now also examined the thermostability of the POL γ A in absence or presence of POL γ B. In short, we found that mPol γ A was more stable than hPOL γ A when in complex with the accessory subunit, regardless of if it was in complex with mPol γ B or hPOL γ B (**Supplementary Fig. 12**). In this case, we believe that T525 and R529 (in mPol γ A) are responsible for the increased interaction between the L-helix and POL γ B, which in part, also may affect the transition between R and E conformers.*

*With that said, it is likely that one or several of the non-conserved residues in POL γ B are also responsible for the different conformations, especially since switching the accessory factors by making chimeric enzymes has a clear effect on the catalytic rate (**Fig. 4h, j**).*

Referee #1, comment 4. Enzyme kinetics (Fig. 4H): due to the processivity of the polymerase, k_{cat} (V_{max}) for the holoenzymes is dominated by k_{off} of enzyme from DNA, not the catalytic rate. 17-33/min are much slower than the polymerization rate measured by the presteady state rate. Thus, the faster k_{cat} for the mouse Polg may not reflect superior catalytic activity but faster off-rate. It is interesting though the K_M (dNTP) for mPol g is lower than hPolg, but hPolg has tighter affinity to DNA (Fig.S6D), can these be rationalized from the structures?

Our Response

The issue raised by the reviewer is very interesting, but we do not have a definitive answer at this point. We do observe that the conformational changes in the human enzyme during the transition between the elongation and exonuclease states are much larger than those seen in the mouse enzyme. However, due to the lack of detailed structural information on the elongation phase, it is difficult to determine whether additional conformational differences could explain the observed variation in k_{cat} .

*To address the reviewer's question, we have now measured k_{off} for both mouse and human POL γ . We note that mouse POL γ has a slightly higher k_{off} (see **Supplementary Fig. 14b** in the revised manuscript) and exhibits marginally weaker DNA-binding affinity (K_d) (see **Supplementary Fig. 13d-e**, experiment revised from the previous version of the manuscript). Additionally, as noted earlier, we observe a somewhat lower K_m (**Fig. 4j**) for mouse POL γ compared to the human enzyme. However, all these differences are relatively small, and we do not observe any major differences in the processivity assay.*

In conclusion, since k_{off} is slightly higher for the mouse enzyme, this could contribute to a higher k_{cat} , as a faster dissociation rate allows the enzyme to release DNA more readily. One could speculate that these two factors, variation in conformational flexibility and a slightly higher k_{off} , may together contribute to the observed differences between the human and mouse enzymes. At this point, this is just speculations, but it will be interesting to follow up in future studies, when we obtain more detailed, structural information for POL γ and structural transitions during elongation.

Referee #1, comment 5. Authors should speculate the structural basis for chimera holoenzymes, hAmB and mAhB activities. What would the subunit interface for these chimera enzymes look like, more human- or more mouse-like?

Our Response:

*This is an interesting question, and to answer it we decided to solve the structures of the chimera enzymes as well. Interestingly, the mAhB chimera was able to enter both a human-like and a mouse-like conformation, which suggests that the accessory factor is not solely responsible for the change in conformation. However, the hAmB chimera only entered a mouse-like conformation, although the density of the DNA were poor compared to the structures with mPol γ A. Overall, the hAmB data was more heterogenous and it is possible that the mPol γ B forces the enzyme into the mouse-like E conformer while the hPOL γ A still strives to enter the human-like conformation but is denied by the accessory subunit. Please see **Supplementary Figs. 7–11** for the newly added data.*

Referee #1, comment 6. Molecular weight standard should be included in gels in Figure 2, as well as the lengths of 'T' and 'P' in panel A. Also, information on mtSBB and mTwinkle protein cloning and purification. was missing.

Our Response:

We have now clarified in the figure legend that “Template (T) indicates 4 kb, and the products are around 20 kb, which is beyond the range of the standard.”

To directly address the reviewer's concern, we have repeated the experiments in **Figure 2b** to ensure that the size marker is now clearly visible. Size markers were already included in the previous version of the experiment, but the higher bands were too weak to be displayed in the figure. Radiolabeling of the 1 kb ladder is sometimes inefficient, leading to weak bands, particularly those above 3 kb. In the new experiments, the bands are clearly visible.

For **Figure 2d**, we also used a size marker (can be seen in the source data accompanying the paper), but again the longer bands were difficult to visualize, and the marker is therefore not included in the displayed experiment. However, in **Figure 2d**, the focus is on the dominant negative effect of the Y933C variant, and the individual reactions serve as controls for each other.

Please note that we consistently include size markers on our gels (as seen in the **source data files**). For our rolling-circle experiments, we use a 1 kb ladder, which extends up to 10 kb. However, under optimal conditions, our products exceed 20 kb. Due to gel resolution limitations at these large sizes, it is not possible to determine the exact product length. Nevertheless, the most relevant reference in all of these experiments is the input template (4 kb band), as it is always clearly visible. This is because not all of the input template is utilized by the replisome in the reactions, but it consistently gets labeled by POL γ due to enzyme idling at the 3'-end. This provides a reliable reference point that is easy to follow.

Referee #1, comment 7. Information on mtSSB and mTwinkle protein cloning and purification. was missing.

Our Response:

Please note that throughout the manuscript, we use human mtSSB and human TWINKLE, as these enzymes, together with mouse Poly, form a highly active replisome. In response to the reviewer's comment, we have further clarified this point in the revised manuscript. Additionally, to demonstrate that shifting mtSSB does not affect the outcome of our experiments, we repeated one of the key experiments using mouse mtSSB (please see the new experiment added as **Supplementary Fig. 1b-c** in the revised manuscript).

Similarly, we did not work with mouse Twinkle, as our focus was on comparing the impact of individual Poly mutations, which can be effectively studied using human TWINKLE. The replisome experiment was included as an additional approach to assess Poly activity and was consistent with the data obtained from Poly in isolation and with both mouse and human mtSSB. Given that cloning mouse Twinkle and repeating all experiments would take 6 months, we prioritized experiments directly related to Poly (please see the new and revised experiments in **Figures: 1a-f, 2b,c, Supplementary Fig 1a-c, 2a,c and d, 7-12, 13a-f and 14b**).

In the revised manuscript, we have made the following addition to the text:

On page 7, we now state:

" To minimize variability in our experiments, we consistently used human mtSSB and only varied POL γ between its human and mouse variants. To validate this approach, we repeated the experiment using mouse mtSSB, obtaining similar results (compare Fig. 1b-c with Supplementary Fig. 1b-c)."

On page 8, we now state:

“To minimize variability, human mtSSB and TWINKLE were used in all experiments, with only POL γ varied between its human and mouse variants.”

Referee #1, comment 8. Fig S6D: hPOL γ DNA binding curve does not fit the actual data, data fitting should be re-performed, the K_d for human should be tighter than reported.

Our Response:

Thank you for your comment. We have repeated the experiments since we found that the previous fitting issue was likely due to instability in the interaction between PolyA and PolyB at lower concentrations, which affected DNA binding measurements.

*To address this, we repeated the experiment using a higher PolyB concentration, ensuring that the complex remained intact throughout the measurement. With these conditions, the fitting is now significantly improved, and the revised K_d values more accurately reflect the binding affinity of human and mouse Poly for DNA. The updated data and fitting curves are now included in **Supplementary Fig. 13d-f** of the revised manuscript.*

Referee #1, comment 9. Data in Fig S6G and S7A appear to be inconsistent: in Fig S6G, the hAhB did not produce long product up to 10 min whereas mAmB produced longer product in 5 min and nearly extended all primers to FL in 10 min. However, in Fig. S7A, hPOL γ performed similarly if not better than mPOL γ in 5min. What could be the reason for the discrepancy?

Our Response:

*Thank you for your observation. The apparent discrepancy between **Supplementary Fig. 13g** and **Figure Supplementary Fig. 14a** arises because these experiments were performed under different conditions, making a direct comparison between them difficult.*

***Supplementary Fig. 13g** was conducted in the presence of mtSSB, which stimulates the polymerase activity and removes secondary structures in the template. This experiment was performed with an excess of polymerase and analyzed on a native gel, where it is impossible to determine the exact size of intermediate products unless they are full-length.*

***Supplementary Fig. 14a**, on the other hand, is a processivity experiment, performed with limiting amounts of polymerase (e.g. template in excess) in the absence (lanes 4–7) or present of heparin (allowing only for single-primer extension) (lanes 10-13). This gel is denaturing, allowing us to precisely define product lengths, although longer products may either spread out (become less visible) or become trapped in wells. Additionally, since mtSSB was not included in this assay, DNA synthesis could also be influenced by secondary structures, affecting the extension efficiency.*

For these reasons, these two experiments are not directly comparable. However, across multiple independent experiments in the manuscript, we consistently observe that the k_{cat} (or polymerization speed) of mouse Poly is higher than that of human Poly, even though the difference is less pronounced in the processivity assay.

Referee #1, comment 10. Animal studies more description on animal behaviors.

a. Treadmill was listed in the method, but no result of the test is presented.

Our Response:

Thank you for your observation. We removed the treadmill from the methods as not relevant. It was inserted by mistake in the original submission.

Referee #1, comment 11. Rationalize why PolgA449T/KO is more defective than Polg A449T/A449T

Our Response:

The most likely explanation is a gene-dosage effect. The A449T/A449T homozygous variant retains some catalytic activity from both alleles, whereas the A449T/KO variant has only one functional allele carrying the A449T mutation, with the other allele being nonfunctional. The absence of a second, even partially active allele likely results in a more severe phenotype due to an overall reduction in POL γ A protein levels and enzymatic activity.

Referee #1, comment 12. What would be the basis for the tissue specificity for OXPHOS activity and the complex specificity, i.e., Fig. S11A, Polg A449T/A449T appears cause more damage to CI than CIV in SKM, but equally defective in Liver and have no effect on either in Brain.

Our Response:

The CI activity in the skeletal muscle from A449T/A449T mutants was not significant. We repeated the analysis and increased the number of samples and confirmed that CI activity was indeed normal.

In Supplementary fig. 18

We only analyzed the spectrophotometric activities of those tissues showing a reduction in mtDNA amount (figure 5). Accordingly, panel A refers to the skeletal muscle of A449T/A449T mutants, panel B to the skm of A449T/KO mice, panel C to the liver of W726S/ W726S mice, panel D to the A449T /G826S mice which had reduced mtDNA content in all the analyzed tissues. The tissue-specificity in the respiratory chain defects observed in our mouse models is probably related to signaling and/or metabolic adaptations in the different organs. This is a well-known and largely unexplained finding in numerous mitochondrial diseases and reinforces the relevance of our genetically modified mice as models for polg-related defects.

Referee #1, comment 13. Fig. S12 How do these data compare to the same animals at 24th month?

Our Response:

We only presented the data at 3 months of age for all the mutants, and aged the Y933C animals up to two years. Doing the aging protocol for all the mutants would just be unfeasible. We are aging the A449T/G826S mice but at the moment the oldest animals we have are 1 year old. Their characterization will be the object of a subsequent publication.

Minor Points:

1. Line 80: Reference correction

Our Response: Thank you for your comment. We have now corrected the reference.

2. Unify the use of POLG, Polg, and POL γ

Our Response: Thank you for your comment. We have now standardized the terminology as follows:

- **Human gene:** *POLG* (italicized, all caps)
- **Mouse gene:** *Polg* (italicized, first letter capitalized)
- **Human protein:** POL γ (all caps)
- **Mouse protein:** Poly (first letter capitalized)

3. Figure 4: Panel A missing from the figure legend

Our Response: Thank you for pointing this out. We have now added the missing panel.

4. Line 338: Suggest changing “Based on the mtDNA data” to “Based on the mtDNA copy number”

Our Response: Thank you for your suggestion. We have now made this correction.

5. Unify ‘in vitro’ format

Our Response: Thank you for your comment. We have now reviewed the entire manuscript and ensured consistency in formatting.

6. Line 616: Add a full stop at the end of “... recombination as described in10”

Our Response: Fixed.

7. Line 630: Delete ‘10’

Our Response: Fixed.

8. Line 632: Missing punctuation at the end of “...Figure S8B”

Our Response: Fixed.

9. Superscript ‘32’ in ³²P throughout the text

Our Response: Fixed.

10. Use prime notation for 5' and 3' ends throughout the text

Our Response: Fixed.

11. Line 738: Correct temperature format to 37°C

Our Response: Fixed.

12. Line 745: Delete the dash from -3' in the oligo sequence

Our Response: Fixed.

13. Line 751: Change “0,1%” to “0.1%”

Our Response: Fixed.

14. Line 757: Confirm if 42(T) is a typo

Our Response: We confirm that 42(T) is NOT a typo. This is a 70-mer oligonucleotide with 42 T repeats at the 5' end, followed by 28 specific nucleotides, as indicated. TWINKLE needs a fork structure to bind

15. Line 776: Use subscript for MgCl₂

Our Response: Fixed.

16. Lines 798-799: Change KD to K_d

Our Response: Fixed.

Reviewer #2 (Remarks to the Author):

Response to Reviewer #3

We appreciate the insightful comments that have helped improve our manuscript. Below are our responses to each point raised.

Referee #3, comment 1. The in vitro primer extension reactions with mouse DNA Poly have been performed with human mtSSB, and not with the mouse mtSSB. The higher activity of mouse PolgAB on the DNA substrates could be due to the human mtSSB or TWINKLE used in the assays. This needs to be experimentally addressed.

Our Response

*We acknowledge the reviewer's concern regarding the use of human mtSSB in our in vitro primer extension reactions with mouse DNA Poly. To address this point, we have now repeated this key experiment using mouse mtSSB instead of human mtSSB to verify that this substitution does not alter the outcome of our findings. The results confirm that the presence of human mtSSB does not impact the conclusions drawn, and these new data are now included as **Supplementary Fig. 1b-c** in the revised manuscript.*

As clearly stated in the revised version of the manuscript, we have consistently used human mtSSB and human TWINKLE in our assays, as these enzymes, together with mouse Poly, form a highly active replisome. We did not use mouse TWINKLE in our study because our focus was to compare the effects of individual Poly mutations relative to wild-type Poly. This comparison can be effectively performed using human TWINKLE, which provides a robust and well-characterized replisome system. Importantly, all Poly mutations were assessed under identical conditions, with wild-type mouse Poly always used as a reference. Please note, that the inclusion of the replisome experiment was intended as an additional means to assess Poly activity and was found to be consistent with data obtained from Poly in isolation, both with mouse and human mtSSB. Given that cloning mouse Twinkle and repeating all relevant experiments would require an estimated 6 months, we prioritized experiments directly related to Poly, ensuring that our main findings remain well-supported. The revised

manuscript now includes new and revised experiments (**Figures 1a-f, 2b,c, Supplementary Fig 1a-c, 2a,c and d, 7-12, 13a-f and 14b**) that reinforce our conclusions.

In the revised manuscript, we have made the following addition to the text:

On page 7, we now state:

“To minimize variability in our experiments, we consistently used human mtSSB and only varied POL γ between its human and mouse variants. To validate this approach, we repeated the experiment using mouse mtSSB, obtaining similar results (compare Fig. 1b-c with Supplementary Fig. 1b-c)”

On page 8, we now state:

“To minimize variability, human mtSSB and TWINKLE were used in all experiments, with only POL γ varied between its human and mouse variants.”

Referee #3, comment 2. In general the mouse PolgAB mutants have slightly higher activities than human PolgAB. The study does not nail down why this is the case. PolgB is a processivity factor. Does mouse PolgAB have higher processivity or rate of DNA synthesis or lower exonuclease activity? Such mechanistic analyses or discussion is missing here

Our response: Please see our response after comment 3

Referee #3, comment 3. The velocity in figure 4F reported for the mouse PolgAB is higher. This could be due to better assembly of the mouse complex on the DNA substrate rather than higher processivity or rate of DNA synthesis.

Our Response to point 2 and 3:

The reviewer raises two very important issues, which has occupied us for a while. The higher activity observed for mouse Pol γ AB compared to human Pol γ AB and we would like to have a deeper mechanistic explanation for this effect.

*We examined processivity, but our assays did not indicate major differences between human POL γ and mouse Poly. We also analyzed the exonuclease activity of human POL γ and mouse Poly (**Supplementary Fig. 13a-c**) and found it to be comparable between the two enzymes.*

*We did observe a higher DNA synthesis rate (k_{cat}) for the mouse Poly enzyme (**Figure 4h-j**). We have reworked the figure (now Fig. 4h-j) and normalized the rates against active enzyme-DNA complex and present only K_m and k_{cat} . Both mouse and human POL γ AB displayed similar ability to assemble on the DNA (~95% of DNA was bound to enzyme, data presented in source data), and we conclude that the incorporation rate is in fact faster in mouse. We discuss this finding in our response to **reviewer 1, point #4**. Please see our response above, for an in depth discussion. As outlined in this response, we have also included new data for k_{off} measurements, which provide more mechanistic insights.*

Taken together, our data indicate that the enhanced activity of mouse Poly is likely due to a combination of structural flexibility, a slightly higher k_{off} , and a subtle difference in DNA-binding affinity. At this point, these remain hypotheses, but future structural studies on POL γ and its conformational transitions during elongation will provide further insights. We have

now expanded the discussion to reflect these mechanistic considerations in the revised manuscript.

In the revised manuscript, we have made the following addition to the text in the discussion:

On page 22, we now state:

***“Our cryo-EM studies reveal structural differences between the human and mouse enzymes in the error-editing state. In mice, the distal POL γ B subunit and the AID subdomain exhibit less internal motion during the transition from replication to error-editing mode, which provides a structural basis for the milder phenotypic effects observed in mouse models. Consistent with this, Yin and colleagues have shown that the distinct functions of human POL γ B are attributed to its dimeric structure, where the monomer closest to POL γ A enhances DNA interaction, while the distal monomer increases the reaction rate²⁷. Our findings extend this model by demonstrating that species-specific conformational differences influence polymerase efficiency and disease mutation tolerance.*”**

***This structural rigidity may also explain why the A449T mutation near the AID domain leads to a milder phenotype in mice compared to humans. Additionally, the reduced flexibility of the mouse holoenzyme appears to enhance nucleotide incorporation efficiency, as POL γ B plays a critical role in stabilizing POL γ A, improving DNA-template engagement, and accelerating catalytic turnover. Indeed, our kinetic measurements (Fig. 4h–j) show that even subtle repositioning of POL γ B can significantly enhance polymerase activity, a mechanism that could broadly counteract the functional deficits caused by diverse POL γ A mutations. Given the clinical heterogeneity of pathogenic POL γ A variants, targeting POL γ B could be a promising therapeutic strategy for restoring polymerase function across multiple disease-associated mutations.*”**

Referee #3, comment 4. The differences in the exonuclease activity of mutant mouse Poly and human Poly could be responsible for the different activity which is not compared here.

Our Response:

*To address the reviewer’s question, we have now performed exonuclease activity experiments and carefully quantified the results (see **Supplementary Fig.13a-c**). Our data show that the exonuclease activity of mouse and human POL γ is very similar, with mouse POL γ exhibiting slightly higher activity. However, this difference is minimal and in the opposite direction of what would be expected if exonuclease activity were solely responsible for the observed differences in polymerase activity.*

Referee #3, comment 5. From the gels in Figures 1 and 2, it appears that using 1:1 ratio of wild-type Poly and mutants recovers the defects produced by the mutants alone. Interestingly, this appears to be the case for both recessive and dominant mutants. What are the mechanistic implications of this observation? Increasing the proportion of the Y955C (or mouse Y933C) mutant from 1:1 to 5:1 reduces the DNA synthesis. How would the recessive mutants fair under these conditions?

Our Response:

Thank you for your interesting question. We have now performed experiments for both the mouse and human variants and included a condition where wild-type (WT) Pol γ was added in 5-fold excess as a control (Supplementary Fig. 2c-d).

WT Pol γ was already in excess in these experiments, and increasing it further (5-fold excess) did not enhance DNA synthesis. As expected, adding a five-fold excess of the recessive A467T and W748S variants did not impact WT replisome function.

Interestingly, both human and mouse versions of G848S/G826S exhibited a slight dominant-negative effect on the WT replisome, despite being classified as recessive. This suggests that G848S may have a greater functional impact than A467T or W748S. Consistent with this, no reported homozygous G848S cases have survived beyond three months, and combining G848S with A467T or W748S results in a more severe phenotype than A467T/W748S alone.

We have now included this observation in the manuscript. However, further experiments are needed to fully understand this effect. Notably, this aligns with clinical observations of late-onset mitochondrial disease in heterozygous POLG mutation carriers (personal communication with Rita Horvath and the POLG Foundation), suggesting a potential avenue for future research.

In the discussion we now write:

“Furthermore, while the dominant Y933C mutant interferes with WT protein function, the recessive mutants do not exert this effect. Interestingly, the G848S mutant exhibits a slight negatively dominant effect in vitro, raising the question of whether carriers of this mutation might develop a very mild mitochondrial phenotype later in life.”

Referee #3, comment 6. Line 118: Related to the claim “Using cryoelectron microscopy (cryo-EM), we establish the structure basis of modulating Pol γ activity by disease causing mutations in mouse”.

Cryo-EM structures are used to understand the interactions between Pol γ A and Pol γ B in mouse and human proteins to explain the differences observed between the polymerase activities in the two models. Structures of human (or mouse) Pol γ A mutants under focus have not been presented. Moreover, except for the mutant A449T (A467T in human), none of the other mutants have been discussed in relation to the cryo-EM structures presented in the manuscript. Thus, the claim to the structural basis of the activities of the disease-causing mutations is not well substantiated.

Our Response: We thank the reviewer for pointing this out and agree that the formulation was not accurate. We have corrected this in the manuscript.

Referee #3, comment 7. Supplementary Fig. S6: There are two bands representing Pol γ Bound to Primer-template, specially at the higher protein concentrations. Could the authors speculate the origin of the two bands?

Our Response:

Thank you for your observation regarding the two bands in Supplementary Fig. 6 (Supplementary Fig. 13d in the new version of the manuscript). Our interpretation is that this is an artifact caused by the very high protein concentration and incomplete saturation of the

POL γ complex. At these high concentrations, some free POL γ A remains unbound and can associate with the POL γ holoenzyme (composed of one POL γ A and a POL γ B dimer) already bound to the template, leading to the observed supershift. However, this interaction is not stable, as it disappears upon adding more template or increasing POL γ B levels, with the one-to-one POL γ -template complexes remaining bound.

*We have repeated the EMSA using a higher concentration of POL γ B while avoiding excessively high POL γ levels. Under these conditions, the supershift was no longer observed, and the new K_d measurement is significantly improved (**Supplementary Fig. 13d-f** in the revised manuscript).*

Referee #3, comment 8. Supplementary Fig. S1. “Mutations impair polymerase activity of the human POL γ across all dNTP concentrations tested”. The title is incorrect as different lanes show a time-course at a constant dNTP concentration.

Our Response: Thanks, the title is now corrected.

Response to Reviewer #5

We thank the reviewer for their encouraging remarks and valuable suggestions. Please find our responses below.

Referee #5, comment 1. These mutations are located in different regions of the POL γ A protein. The A467T mutation is situated in the thumb region of the polymerase domain, while the W748S mutation is in the linker region that connects the proofreading and polymerase domains, and is important for interactions between POL γ A and POL γ B.” Would be nice to have a schematic showing the position of these mutations on the protein structure

Response:

*We agree with the reviewer that a schematic representation would be valuable. We have now included a figure illustrating the positions of these mutations on the POL γ A protein structure to provide a clearer visualization of their locations (See **Fig. 3a** in the revised version of the manuscript)*

Referee #5, comment 2. Figure 1 and Figure 2

It is true there is a milder effect of some mutations over others. But given the small differences, to make a proper comparison among mutants, an SDS page before dilution to final concentration should be provided showing that comparable amount of protein was used in the assay. It would also be good to have an idea of how these mutations affect the overall stability of the protein, maybe by looking at the thermal stability.

Our Response:

We agree with the reviewer that ensuring comparable protein concentrations is crucial. As part of our standard procedure, we always carefully determine protein concentrations for each new batch, using a wild-type (WT) batch as a reference. The WT concentration is

measured via UV absorbance, and a standard curve is used to quantify all protein variants. If the proteins have been stored for more than three months before experiments, we routinely re-verify their concentrations. Please also see our response to reviewer 1, point #1, in which we discuss specific problems related to the mouse W726S mutant protein.

To ensure that comparable protein levels were used across all experiments, we confirmed the measured protein concentrations by separating and visualizing the POLG variants on an SDS-PAGE gel (see **Supplementary Fig. 1a**).

In the revised manuscript, we have made the following addition to the text:

On page 7, we now state:

“All proteins were expressed in recombinant form, and their concentrations were determined before enzymatic assays (Supplementary Fig. 1a). Unless otherwise stated, all wild-type and mutant variants (both human and mouse) were used at equal concentrations in the biochemical experiments.”

We also agree with the reviewer that it is valuable to investigate how these mutations affect the overall stability of the protein. To address this, we performed thermal stability analyses and found that in the absence of POL γ B, all POL γ A variants exhibited similar stability to WT. However, in the presence of POL γ B, all mutants showed slightly reduced stability, except Y933C, which remained comparable to WT. While these differences are relatively small, we cannot exclude the possibility that reduced stability is part of the phenotype of these mutants. These results have now been included as a new figure in the revised manuscript (**Supplementary Fig. 12**).

Referee #5, comment 3. Clearly, mPol γ B contributes to the differences in activity observed between the human and mouse POL γ holoenzymes. The effects of mutations are similar for hPOL γ A and mPol γ A in isolation, but when combined with their accessory factors, the stronger stimulatory effect of mPol γ B partially conceals the negative effects of disease-causing mutations in the mouse compared to the human system”

It is clear how POL γ B stimulates DNA synthesis, and mPol γ B has a stronger effect than the human counterpart. The structural observations suggest that the more flexible human proteins (within AID and POL γ B domains) are the reason for this difference. To strengthen this hypothesis a set of complementary DNA binding assays would be useful. How does the affinity for the DNA changes in presence and absence of POL γ B? Does mPol γ B increase the affinity of hPOL γ A for DNA?

Our Response:

We appreciate the reviewer’s insightful suggestion regarding complementary DNA binding assays. While we recognize the importance of understanding how POL γ B influences DNA affinity, our biochemical data did not reveal a substantial difference in DNA binding between the human and mouse enzymes. Based on this, we did not perform additional DNA binding experiments, as we focused on the functional and structural differences that more prominently explain the distinct effects of disease mutations in human and mouse models.

Our cryo-EM studies reveal key structural differences between the human and mouse enzymes in the error-editing state, particularly in the motion of the distal POL γ B subunit and the AID

subdomain. These observations suggest that the reduced flexibility of the mouse holoenzyme enhances nucleotide incorporation efficiency and partially compensates for the effects of disease mutations. We have discussed this in detail in the manuscript, and our findings support the hypothesis that species-specific conformational differences contribute to polymerase efficiency and disease mutation tolerance.

In the revised manuscript, we have made the following addition to the text in the discussion:

On page 22, we now state:

“Our cryo-EM studies reveal structural differences between the human and mouse enzymes in the error-editing state. In mice, the distal POL γ B subunit and the AID subdomain exhibit less internal motion during the transition from replication to error-editing mode, which provides a structural basis for the milder phenotypic effects observed in mouse models. Consistent with this, Yin and colleagues have shown that the distinct functions of human POL γ B are attributed to its dimeric structure, where the monomer closest to POL γ A enhances DNA interaction, while the distal monomer increases the reaction rate²⁷. Our findings extend this model by demonstrating that species-specific conformational differences influence polymerase efficiency and disease mutation tolerance.

This structural rigidity may also explain why the A449T mutation near the AID domain leads to a milder phenotype in mice compared to humans. Additionally, the reduced flexibility of the mouse holoenzyme appears to enhance nucleotide incorporation efficiency, as POL γ B plays a critical role in stabilizing POL γ A, improving DNA-template engagement, and accelerating catalytic turnover. Indeed, our kinetic measurements (Fig. 4h–j) show that even subtle repositioning of POL γ B can significantly enhance polymerase activity, a mechanism that could broadly counteract the functional deficits caused by diverse POL γ A mutations. Given the clinical heterogeneity of pathogenic POL γ A variants, targeting POL γ B could be a promising therapeutic strategy for restoring polymerase function across multiple disease-associated mutations.”

Referee #5, comment 4. In the EMSA shown in figure S6.C there is a double shift at high concentration. The reason for this was not explained in the text. A different assay like fluorescence anisotropy with a labelled DNA template could be more suitable in picking up small differences in affinity to the DNA.

Our Response:

Thank you for your observation regarding the two bands in Supplementary Fig. 6). Our interpretation is that this is an artifact caused by the very high protein concentration and incomplete saturation of the POL γ complex. At these high concentrations, some free POL γ A remains unbound and can associate with the POL γ holoenzyme (composed of one POL γ A and a POL γ B dimer) already bound to the template, leading to the observed supershift. However, this interaction is not stable, as it disappears upon adding more template or increasing POL γ B levels, with the one-to-one POL γ -template complexes remaining bound.

We have repeated the EMSA using a higher concentration of POL γ B while avoiding excessively high POL γ levels. Under these conditions, the supershift was no longer observed,

and the new K_d measurement is significantly improved (**Supplementary Fig.13d-f** in the revised manuscript).

Referee #5, comment 5. Figure S7B:

Although the mutations have different numbers between human and mouse, it would still be visually helpful to have the “h” or “m” in Figure S7B in front of the mutated residues

Our Response: We added as reviewer suggested. It is Supplementary Fig. 14 in the new version of the manuscript.

Referee #5, comment 6. W748S>G848S>A467T>Y933C

As the mutations with higher severity are on the right, the symbol should be inverted: < instead of >

Our Response: We changed as reviewer suggested.

Referee #5, comment 7. Taken together, our findings point to POL γ B as a potential therapeutic target, which could be leveraged to mitigate the effects of various POL γ mutations.” While it is clear how the POL γ B subunit enhances the polymerase activity, remains undiscussed how this subunit could be used as a potential therapeutic target

Our Response: We thank the reviewer for this comment. Please see our response to your comment 3

**Responses to referees' comments – Nature comm. manuscript 24-53797:
"Modelling POLG mutations in mice unravels a critical role of POL γ B in regulating phenotypic severity"**

Response to Reviewer #1

The authors have thoroughly addressed our critiques. The revised manuscript include new data on thermostability measurements of Pol gamma variants, two new cryo-EM structures and repeated enzymatic analyses. We are satisfied with the revision and believed its has met the requirement for publication.

Our Response:

We are pleased that the reviewer is satisfied with the changes made to the manuscript and we appreciate the additional minor comments that have helped improve the manuscript further. Below are our responses to each point raised.

Referee #1, comment 1. Reference #40 appears to be incorrect.

Our Response:

Thank you for noting this. We have corrected the reference.

Referee #1, comment 2. Double check the Kd value presented in Sup Fig.13f, which is three-orders of magnitudes higher than the published value.

Our Response:

As noted by the reviewer, the observed Kd values are indeed higher than those reported previously by others and us (Lim et al., J Mol Biol. 2003; Bratic et al., Nat Commun. 2015). The lower apparent Kd values in Sup Fig. 13f are due to differences in experimental design and buffer conditions used here, compared to previous studies.

In our current work, we used a large excess of POL γ B to ensure that the POL γ A–POL γ B complex remains intact during the assay. This precaution was taken as the POL γ complex dissociates (to free POL γ A and POL γ B) at high dilutions (Young et al., HMG 2011), which would result in a lower concentration of complex than expected and thus bias the determination of apparent Kd to higher values.

In addition, we used a relatively low NaCl concentration (30 mM NaCl), which favors DNA binding. We also included ddCTP and Mg²⁺ to prevent exonuclease activity, whereas the previous EMSA experiments were performed in the absence of nucleotides and Mg²⁺ (Lim et al., J Mol Biol. 2003; Bratic et al., Nat Commun. 2015).

Due to the tight binding under these conditions, fitting the data using a classical binding isotherm (“Binding ratio = $B_{max} \times [POL\gamma \text{ complex}]_{free} / (Kd + [POL\gamma \text{ complex}]_{free})$ ”) did not yield a satisfactory fit, as the assumption that $[POL\gamma \text{ complex}]_{free} \approx [POL\gamma \text{ complex}]_{total}$ does not hold true. Instead, we employed Morrison’s quadratic equation for tight binding, which better reflects the binding behavior at the high-affinity limit and low ligand concentrations. In conclusion, Kd is a condition-dependent parameter, and since the conditions used here are quite different from those used in some previous studies, the Kd values are much lower.

Please note, when under similar conditions to those used here, others and us obtain Kd values that are quite similar (Yakubovskaya et al., JBC 2006; Valenzuela et al., Nature 2025).

Response to Reviewer #3

Authors have satisfactorily addressed all the comments and concerns raised, improving the manuscript considerably. I have some minor comments:

Our Response:

We are pleased that the reviewer is satisfied with the changes made to the manuscript, and we appreciate the additional minor comments that have helped further improve the manuscript. Below are our detailed responses to each point raised

Referee #3, comment 1. Line 198: “This observation aligns with the fact that dominant mutations are relatively mild, leading to mtDNA deletions rather than depletion in affected patients”. Please cite relevant references supporting the statement.

Our Response:

Thank you for your comment. We have now added a reference to support this statement in the revised manuscript.

Referee #3, comment 2. Line 113: “Both mouse PolyA and PolyB are conserved in most mammals, with mPolyA sharing 85% sequence identity with its human counterpart”. I think the authors want to convey that both PolyA and PolyB are conserved among most mammals, rather than ‘mouse’ PolyA and PolyB are conserved.

Our Response: Thank you for pointing this out. We have changed to “PolyA and PolyB are conserved among most mammals, with mPolyA sharing 85% sequence identity with its human counterpart”

Referee #3, comment 3. Authors should note in the methods that the processivity experiments were performed without mtSSB to dissuade the readers from comparing DNA syntheses in Figure 1b and Supplementary Figure 14a.

Our Response: Thank you for pointing this out. We have added a note in the methods, processivity section,” Please note that the processivity assay were performed without mtSSB in the reaction, which differ from our DNA synthesis assay.”

Referee #3, comment 4. Line 210: “To better understand why mutations generally cause a less severe molecular phenotype in mouse vs human Pol γ , we employed single-particle cryogenic electron microscopy to determine the structure of the mPol γ WT-DNA complex”. As no cryoEM structures of the mutants are reported, the structures presented inform more on the differences between mPol γ WT and hPOL γ WT rather than explaining less severe phenotypes observed in mouse Pol γ mutants.

Our Response: Thank you for pointing this out. We have changed to “To better understand the differences between mouse and human Pol γ , we employed single-particle cryogenic electron microscopy to determine the structure of the mPol γ WT-DNA complex.”

Referee #3, comment 5. Figure 6. It will be helpful to add H & E, COX/SDH etc. on the figure panels.

Our Response: We have modified the figure as requested.

Referee #3, comment 6. Error bars in Figures 1e, 1f, 4h, 4i and 4j should be defined in the respective figure legends.

Our Response: Thank you for pointing this out. We now added in Figure legends:” the error bars present the standard deviation” in Fig1 e-f and : ”Data are presented as mean \pm s.e.m.” in fig 4h-i, and “Data are presented as mean \pm s.d.” in fig 4j.

Referee #3, comment 7. Line 779: “The template used for the steady state kinetics is the same that was used in the DNA synthesis assay on ssDNA but without 5'-radiolabeling”. There are two templates that were used for DNA synthesis on ssDNA (M13 ssDNA and single-stranded pBluescript SK(+) OriL). Specify which one was used.

Our Response: Thank you for pointing this out. Now we changed to “The template used for the steady-state kinetics assay is the M13 ssDNA template, the same as that used in the DNA synthesis assay on ssDNA but without 5'-radiolabeling.”